# Combined effect of physico-chemical and microbial quality of breeding habitat water on oviposition of malarial vector *Anopheles subpictus*

**Madhurima Seal, Soumendranath Chatterjee**[ID]*

Department of Zoology, Parasitology and Microbiology Research Laboratory, The University of Burdwan, Burdwan, West Bengal, India

* soumen.microbiology@gmail.com

**Data Availability Statement:** All relevant data are within the paper and its Supporting information files.

## Abstract

Mosquitoes prefer diverse water bodies for egg laying and larval survival. Present study was performed with an objective to characterize physico-chemical properties and microbial profiling of breeding habitat water bodies of *Anopheles subpictus* mosquitoes. A field survey was accomplished to check the presence of *An. subpictus* larvae to record per dip larval density in various breeding habitats throughout the year. Physico-chemical and bacteriological properties in relation to mosquito oviposition were assessed. Dissolved oxygen content, pH and alkalinity were found to have major impacts and ponderosity on the prevalence of *An. subpictus* larvae. Larval density showed significant positive correlation with dissolved oxygen content of water and significant negative correlation with pH and alkalinity of habitat water. Comparatively higher population (cfu/mL) of *Bacillus* spp. competent with starch hydrolyzing and nitrate reducing properties were recorded all the breeding habitat water bodies of *An. subpictus*. Higher amplitude of anopheline larvae was portrayed during monsoon and post-monsoon season in clear water with an inclining trend to high dissolved oxygen content and neutral pH. *B. cereus*, *B. megaterium*, *B. subtilis* and *B. tequilensis* prevalent in all habitat water bodies were marked as oviposition attractants of gravid *An. subpictus* mosquitoes. Microbial population played key roles in the modulation of physico-chemical parameters of habitat water with a view to enhance its acceptability by gravid mosquitoes in relation to their oviposition. Better understanding of the interactions along with the control of oviposition attractant bacterial strains from mosquito breeding habitats might contribute to the vector management programme.

## Introduction

Malaria is a serious vector borne disease which is responsible for millions of deaths in tropical and subtropical countries [1]. Females of different species of *Anopheles* mosquitoes serve as vectors of malaria parasites due to their blood sucking behavior [2]. Among different species of *Anopheles*, *An. subpictus* is reported as one of the important vectors of malarial diseases from different parts of the world [3–6]. Breeding of *An. subpictus* occurs in a variety of habitats,

**Funding:** The author(s) received no specific funding for this work.

**Competing interests:** The authors have declared that no competing interests exist.

including stagnant or flowing water bodies having clear or turbid water, brackish or fresh water, water bodies of ponds, lake, polluted water of cement cisterns and submerged rice-fields etc. [7, 8]. Besides human habitations, adult forms of *An. subpictus* frequently occur in cattle sheds [9, 10]. Prevalence of these mosquito vectors depends upon availability of suitable breeding habitats. Different species of mosquitoes have been reported to prefer habitat water with diverse physico-chemical characteristics for their egg laying and larval survival [11, 12]. Even the same species of mosquitoes from different geographic regions have also been found to favour dissimilar type of water bodies [13]. A recent malarial resurgence was reported from different rural and urban areas of Hooghly district [14]. *Plasmodium falciparum* cases were maily reported from urban areas, whereas *Plasmodium vivax* cases were reported from rural areas of Hooghly district [14]. Another study by Amitabha and co-workers reported annual parasitic index (API) for malarial cases were highest in Pandua block of Hooghly district [15].

Numerous research studies have indicated that physico-chemical parameters of larval habitat water have great influence in several life history stages of mosquito vectors including larval survival, time of development and adult fitness [16, 17]. Among these physico-chemical parameters dissolved oxygen content (D.O), alkalinity, pH, turbidity, total dissolved solids (TDS), hardness, conductivity of water and presence of different ions like chloride ($Cl^-$), nitrate ($NO_3^-$), phosphate ($PO_4^{3-}$) etc. are recorded to affect the oviposition of different mosquito species like *An. barbirostris*, *An. gambiae*, *An. vagus*, *An. stephensi*, *An. arabiensis* etc. [18, 19]. Some of these parameters showed strong positive correlation with the larval abundance [20, 21].

In addition to physico-chemical parameters, bacteriological features of water also have significant influence in the modulation of oviposition behaviour of gravid female mosquitoes [22, 23]. The bacterial flora present in the habitat water have the ability to modulate the physico-chemical quality of water [24, 25], thus making it more or less suitable for survival of different mosquito species. On one hand, these breeding habitat bacteria serve as a direct food source for mosquito larvae [26] and on the other hand, these bacteria release some volatiles, which act as chemical attractants for oviposition of gravid female mosquitoes [27, 28]. Several studies indicated that killing of these bacteria by sterilization technique or addition of effective antibiotics to the breeding habitat water led to a reduction in ovipositional response by adult gravid female mosquitoes [29–31].

There is very scanty information available about the breeding and larval habitat characteristics of malarial vector *An. subpictus*. Mosquito control strategy depends basically on larval control. Various strategies applied for mosquito larval control will become much more effective, only when there is complete knowledge about their habitat characteristics. At the same time, it is also essential to identify the biological organisms such as bacteria of habitat water, which produce a preferred environment for the mosquito larvae, because killing or effective management of these organisms will certainly reduce mosquito oviposition and larval survival in the environment, That's why our study is focused whether both biological and chemical breeding habitat parameters have influenced the propagation and multiplication of this particular rural malarial vector species. So, the present study has been aimed to determine the significant physico-chemical characteristics as well as the microbial markers with a special reference to oviposition attractant bacterial strains having a positive influence towards the oviposition behaviour of gravid female *An. subpictus* mosquitoes.

## Materials and methodology

### Study area and study period

The study was conducted in four seasons viz., summer (Mar'17-May'17), monsoon (Jun'17-Aug'17), post-monsoon (Sept'17-Nov'17) and winter (Dec'17-Feb'18) in four blocks

viz., Tarakeswar, Singur, Chinsurah-Mogra and Panduah of Hooghly district West Bengal, India (23˚ 01´ 20" N to 22˚ 39´32" N and 87˚30´ 20" E to 88˚ 30´ 15" E). This district is located at the sea level and has a tropical wet and dry climate. Average annual temperature of this district is 30.44˚C and it receives average 82.25 mm rain fall annually.

## Field survey & collection of habitat water

Suspected water bodies (ponds, drains and rice-fields) of the study areas were checked randomly for the presence of *Anopheles subpictus* larvae. Samples were collected from those water bodies, where larval prevalence of *An. subpictus* were recorded during the study period (Fig 1). No invasive test data was collected during field survey. Only the mosquito larval prevalence was recorded in the water bodies without hampering the normal aquatic flora and fauna of the breeding habitats surveyed. So, no permits were necessary for the field survey and water collection. Twenty replicas were taken per aquatic body type. Twenty five dips were taken with a standard dipper of 250 mL capacity where *An. subpictus* larvae were prevalent. Per dip larval density of *An. subpictus* in each breeding habitat was recorded. Absence of any larvae in a waterbody even after 25 dips was considered as negative. Same mosquito breeding habitats were used for physico-chemical and bacterial sampling from where the larval collection was done, as bacterial composition definitely interferes with the oviposition of gravid female mosquitoes. Water samples from breeding habitats were collected in separate sterile bottles and brought to the Parasitology & Microbiology Research Laboratory, Department of Zoology, The University of Burdwan for both physico-chemical and bacteriological analyses.

## Analysis of physico-chemical parameters of water

Some of the parameters like pH, temperature, total dissolved solids (TDS) and electrical conductivity (E.C) of water were estimated at the time of collection by hand held pH meter, temperature/tds meter and conductivity meter respectively. For estimation of dissolved oxygen (D.O) content, water sample was collected in a stoppered bottle and fixed. Later the D.O content of water was measured in the laboratory by titrimetric method following standard

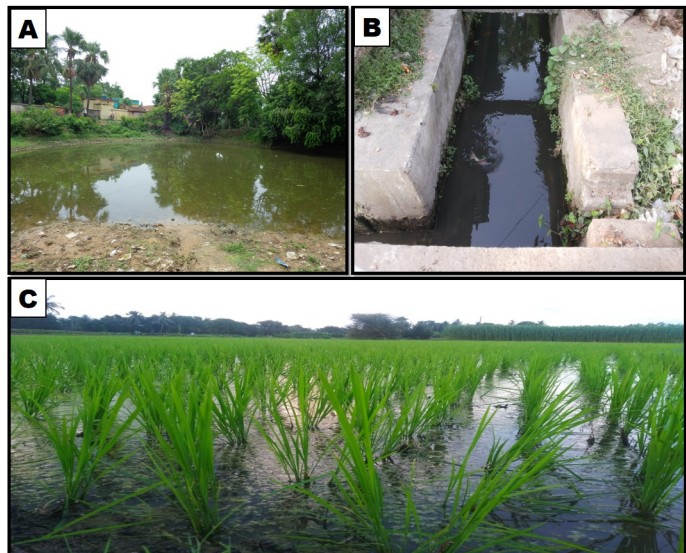

**Fig 1. Different types of aquatic bodies surveyed during the study period.** A: pond water, B: drain water, C: rice-field water.

protocol [32]. Turbidity of the water was measured by digital turbidity meter (LABARD, model no: LIM221) in the laboratory. Other physico-chemical parameters like alkalinity, hardness and concentration of chloride ($Cl^-$), nitrate ($NO_3^-$) & phosphate ($PO_4^-$) were estimated in the laboratory by titrimetric method following standard protocol [32]. Average values with standard error (mean ± S.E) were calculated.

## Analysis of bacterial populations of water

Abundances of different groups of bacterial populations (aerobic heterotrophic, *Bacillus* group, Gram negative, protein hydrolyzing bacteria, starch hydrolyzing bacteria & nitrate reducing bacteria) in the habitat water were determined as colony forming unit (cfu/mL). Water samples were serially diluted with sterilized distilled water and 20–100 μL of diluted sample water was mixed with 25 mL of sterilized media. Then sample mixed media were poured on sterilized petri plates and incubated in a biological oxygen demand (B.O.D) incubator for 24–48 h for colony formation.

For enumeration of aerobic heterotrophic bacterial population, nutrient agar medium was used. For the enumeration of Gram negative and *Bacillus* group of bacterial populations Mac-Conkey agar and Hicrome *Bacillus* agar media were used respectively. To determine the starch hydrolyzing bacterial population, colonies grown over starch agar plates were flooded with freshly prepared iodine solutions and colonies that exhibited clear zone surrounding the growth, were counted as positive bacterial colony for starch hydrolysis. For the determination of protein hydrolyzing bacterial populations, colonies grown over gelatin agar media were flooded with freshly prepared mercuric chloride ($HgCl_2$) solution and the colonies that exhibited clear zone were counted as positive for protein hydrolysis. To determine nitrate reducing bacterial populations, colonies formed over nitrate agar medium were flooded with α-napthol and sulphanilic acid (1:1) and colonies that turned to pink colour were counted as positive for nitrate reduction. Each experiment was performed in triplicate and average number of colonies and standard error (S.E) were calculated. Then in each case colony forming unit (cfu) was calculated by following formula:

$$\text{cfu} = (\text{Total number of colonies} \times \text{dilution factor})/\text{Volume added}$$

## Processing of water samples for bacterial isolation

The water samples were serially diluted (up to $10^{-3}$ dilution) with sterile distilled water and then 40 μL of each diluted samples were mixed separately with 25 mL of sterile and moderately cooled nutrient agar media (Peptone: Beef extract: NaCl: Agar at 5:3:3:18 g/l) and then plated on separate sterile petri plates and incubated in a B.O.D incubator at 32±1ºC for 24–48 h to obtain isolated colonies. To obtain pure culture of bacteria, quadrant streaking technique was employed. Pure cultures of bacteria were maintained on sterile agar slants and stored in refrigerator for further characterizations.

## Phenotypic characterizations of bacterial isolates

Colony characters (size, shape, colour, opacity, elevation, margin etc.) of the bacterial isolates on nutrient agar media were studied following standard methodologies [33–35]. Gram's staining was carried out to observe the shape and Gram characters of the vegetative cells of the bacterial isolates and endospore staining was performed with malachite green followed by counter staining with safranin to detect the presence of bacterial endospores.

## Bio-chemical characterizations of bacterial isolates

Different biochemical properties of bacterial isolates such as production of catalase, indole, methyl red test, vogues proskauer test, citrate utilization, nitrate reduction, urease production, oxidase tests and production of extracellular enzymes like amylase (starch hydrolysis test), lipase (fat hydrolysis test), gelatinase (protein hydrolysis test) were performed following standard methodology [36]. Motility of the bacterial strains were recorded in sulfide indole motility agar medium.

## Ovipositional bioassay

**Mosquito rearing.** Anopheline larvae were collected from different natural breeding habitats in rural areas of Hooghly district with the help of standard dipper of 250 mL capacity [37] and brought to the Parasitology & Microbiology Research Laboratory, The University of Burdwan in live condition. In the laboratory, the mosquito larvae were maintained in plastic trays kept in mosquito cages containing natural breeding habitat water at 28±2ºC temperature and 75±5% relative humidity until the pupa formation. The larvae were fed with Brewer's yeast, algae collected from pond water and dog biscuits in the ratio of 3:1:1 [38]. Pupae were transferred in another white coloured plastic cups of 250 mL capacity containing natural breeding habitat water and the cups were placed in mosquito rearing cages until the emergence of adult mosquitoes. The whole experimental setup was maintained at 28±2ºC temperature and 75±5% relative humidity in an environmental chamber. After emergence of adult mosquitoes, they were identified following the standard keys [7]. The adult *Anopheles subpictus* mosquitoes were offered 10% sucrose solution in cotton pads and allowed to mate freely. Then 3–5 days old females were given blood meal in order to mature their eggs. Then fully gravid female *An. subpictus* mosquitoes were selected and separated in another cage for further ovipositional bioassay.

**Setting up cages to study mosquito oviposition.** Ovipositional bioassay was conducted to evaluate the ability of resident bacterial isolates, which were prevalent all through the year in the natural breeding habitat water bodies of *An. subpictus* mosquitoes to modulate the ovipositional behavior of adult gravid *An. subpictus*. The purified bacterial colonies were inoculated separately on 100 mL of sterilized nutrient broth media and incubated at 32±1ºC in a B.O.D shaker incubator for 16–18 h. One uninoculed nutrient broth was kept for control. Ten adult gravid females *An. subpictus* were released in a mosquito raring cage (30 cm × 30 cm × 30 cm) and they were offered dual choice for oviposition in a cage. In each cage two oviposition cups were placed diagonally at a distance of 21 cm. One cup having 95 mL of sterile distilled water with 5 mL of one type of bacterial suspension which served as a test cup and another cup containing 100 mL of sterile distilled water which served as a control. In another cage, one cup was kept having 95 mL of sterile distilled water + 5 mL mixed suspension of all common bacterial isolates. Five replications were done for each of the tests. The whole experimental set up was maintained in an environmental chamber at 28±2ºC temperature and 75±5% relative humidity at 12:12 h (light: dark) photoperiod in the Parasitology and Microbiology Research Laboratory, Department of Zoology, The University of Burdwan. The number of eggs laid in different test cups and control cups were recorded on the next two consecutive days.

The oviposition activity index (OAI) was calculated using the following formula [39].

Oviposition Activity Index (OAI) = [number of eggs laid in test cups (NT) − number of eggs in control cups (NC)]/ [number of eggs laid in test cups (NT) + number of eggs in control cups (NC)].

## Molecular characterization and phylogenetic analysis of bacterial isolates

Bacterial isolates towards which gravid *An. subpictus* mosquitoes showed significant oviposition attractancy were further selected for molecular analysis. The bacterial isolates were streaked on separate sterilized nutrient agar plates and incubated at 32±1ºC for 24 h in a B.O.D incubator to obtain isolated colonies. On the next day, liquid culture of bacterial isolates were prepared by inoculating isolated colonies in separate sterile nutrient broth medium and incubating at 30±1ºC for 24 h in a B.O.D shaker incubator. The genomic DNA of bacteria was extracted following standard protocol [40]. 1.8 mL of each bacterial broth was taken in separate sterile centrifuge tube of 2 mL capacity and centrifuged at 10,000× for 30 sec at room temperature to obtain the bacterial pellets. Then the genomic DNA of the bacterial isolates were isolated by DNeasy Ultra Clean Microbial Kit (Qiagen) from the respective pellets of the bacterial isolates. After that ~1.5 kb rDNA fragment of bacterial genomic DNA were amplified using 27F (5′AGAGTTTGATCATGGCTCAG 3′) forward and 1492R (5′GGT TAC CTT GTT ACG ACTT3′) primer by polymerase chain reaction (one cycle at 94˚C for 5 min, then at 94˚C for 5 min, thirty five cycles at 58˚C for 1 min, 72˚C for 1 min and then for 7 min) and PCR products were purified using MinElute PCR purification kit (Qiagen). Agarose gel electrophoresis was done to visualize PCR purified DNA products and the PCR products were sequenced bi-directionally by DNA sequencer using universal bacterial forward and reverse primer. Sequenced data were aligned and analyzed by MEGA X software [41]. Phylogenetic trees of the bacterial isolates were created by neighbour-joining (NJ) and maximum likelihood (ML) method [42].

## Scanning electron microscopy of oviposition attractant bacterial isolates

Surface structure of vegetative cells and bacterial endospores were observed through scanning electron microscope. For vegetative cells, a thin smear from freshly prepared liquid bacterial cultures were prepared over cover glasses, whereas for endospores, smear was prepared over cover glasses from 3–4 weeks old bacterial culture. Then the bacterial smear was dried by gently passing them over flame for 4–5 times. Then the cells were fixed chemically in 2.5% glutaraldehyde solution for 45 min and after that they were gradually dehydrated through graded alcohol (30%, 50%, 70%, 90%, 100%) for 5–7 min in each. Finally, they were transferred in iso-amyl alcohol for final dehydration for 5 min. Then the cells were dried in air, coated with gold particle and scanned through scanning electron microscope (Sigma 300, ZEISS).

## Carbohydrate fermentation test of oviposition attractant bacterial isolates

Release of oviposition attractant volatiles is associated with bacterial fermentation of different carbohydrate sources. So, the ability of the bacterial isolates to ferment twenty different carbohydrate sources like Galactose (Ga), Inositol (IS), Mannitol (Mn), Arabinose (Ar), Cellobiose (Ce), Trehalose (Te), Dulcitol (Du), Sucrose (Su), Raffinose (Rf), Dextrose (De), Melibiose (Mb), Sorbitol (Sb), Xylose (Xy), Fructose (Fc), Rhamnose (Rh), Lactose (La), Adonitol (Ad), Mannose (Mo), Salicin (Sa), Inulin (In) were examined over phenol red agar media using sugar fermentation disc (25 mg/disc).

## Antibiotic sensitivity test of oviposition attractant bacterial isolates

Sensitivity of bacterial isolates to twenty different types of commercially available standard antibiotic disc like Kanamycin (30 µg/disc), Bacitracin (10 µg/disc), Amoxicillin (10 µg/disc), Nalidixic acid (30 µg/disc), Ampicillin (10 µg/disc), Penicillin (10 µg/disc), Chloramphenicol (30 µg/disc), Levofloxacin (5 µg/disc), Gentamicin (50 µg/disc), Neomycin (30 µg/disc),

Ofloxacin (5 μg/disc), Norfloxacin (10 μg/disc), Tetracycline (30 μg/disc), Ciprofloxacin (5 μg/disc), Vancomycin (30 μg/disc), Rifampicin (5 μg/disc), Azithromycin (30 μg/disc), Erythromycin (15 μg/disc), Streptomycin (10 μg/disc), Doxycycline (30 μg/disc) were checked over mullar-hinton agar plate by disc diffusion method [43]. This test was performed to determine the potent antibiotics which could have a significant antibacterial effect and have the ability to kill or eliminate particular bacterial strains performing as microbial markers for the oviposition of gravid female *An. subpictus* mosquitoes in their natural breeding habitat water bodies.

## Physiological tolerance test of oviposition attractant bacterial isolates

Physiological properties of oviposition attractant bacterial isolates like sodium chloride (NaCl) tolerance, growth at different temperature and pH of the culture media were recorded following standard methodologies [35, 44, 45]. For temperature tolerance test, the bacterial isolates were inoculated separately in sterilized nutrient broth media and were incubated in B.O.D. shaker incubator for 24 h at different temperatures (15˚C, 20˚C, 25˚C, 30˚C, 35˚C, 40˚C, 45˚C). After the incubation period, amount of bacterial growth was recorded by taking optical density (O.D) at 600 nm. For pH tolerance test bacterial isolates were inoculated separately in sterilized nutrient broth having varying pH (5–11) and were incubated in B.O.D. shaker incubator for 24 h at 32±1˚C. After the incubation period, amount of bacterial growth was recorded by taking O.D at 600 nm. NaCl tolerance test was performed by inoculating the bacterial isolates in sterilized nutrient broth with varying NaCl concentration (upto 5%) and bacterial growth was measured by recording O.D at 600 nm after 24 h of incubation period in a B.O.D shaker incubator at 32±1˚C.

## Statistical analysis

Effect of habitat types and season on larval density of *An. subpictus* was assayed by non parametric Friedman test using SPSS 20.0 software. One-way analysis of variance followed by post hoc tukey test was conducted to evaluate significant differences of physico-chemical parameters among habitat type (ponds, drains & rice-fields) during summer, monsoon & post-monsoon seasons and Mann-Whitney test was performed for significant differences in physicochemical parameters between ponds and drains in winter season using GraphPad Prism 9.0.0 software following Zar [46]. Principal Component Analysis (PCA) was conducted to explore the physico-chemical factors that are responsible for variations of larval density in habitat waters in four different seasons (summer, monsoon, post-monsoon & winter) and Pearson's correlation test with Holms-Bonferroni correction was done between larval density and all the physico-chemical parameters of habitat water using PAST 4.03 software. A generalized linear model (GLM) has been constructed with larval density as dependent variable and all physico-chemical parameters (temperature, dissolved oxygen, alkalinity, pH, turbidity, total dissolved solids, total hardness, electrical conductivity, chloride, nitrate and phosphate) as predictors. We used sigma restricted parameterization and type VI sum of square to calculate the coefficient of prediction equation using Statistica 12 software. Paired t test was performed to determine any significant differences in number of eggs laid by gravid *An. subpictus* mosquitoes between control cups and test cups using GraphPad Prism 9.0.0 software [46].

## Results

### Larval density of *Anopheles subpictus*

Among different types of habitats surveyed, pond water was found to harbour much higher density of *An. subpictus* larvae than rice fields and drains in all the four seasons studied and

**Table 1. Seasonwise per-dip larval density (Mean ± S.D) of *Anopheles subpictus* in different breeding habitats (March 2017-February 2018).**

| Habitat types | Summer | Monsoon | Post-monsoon | Winter |
|---|---|---|---|---|
| Pond | 10.05± 1.16 | 15.16± 6.46 | 16.54± 4.42 | 4.02± 1.21 |
| Drain | 0.8± 0.78 | 2.25± 1.30 | 1.12± 0.92 | 0.46± 0.39 |
| Rice-field | 1.07± 0.55 | 1.57± 0.76 | 1.3± 0.61 | 0.00± 0.00 |

the larval prevalence was found higher during post-monsoon months, followed by monsoon and summer, whereas in winter season larval density was recorded to be lowest (Table 1). Details of larval density in different habitat types have been listed in S1 File. During winter season, due to absence of water, the rice field areas could not serve as potential breeding habitats for mosquito larvae. Results of Friedman test revealed that both season and habitat type significantly influenced larval density of *An. subpictus* mosquitoes ($X^2$ (3) = 88.91, p<0.0001) (S1A & S1B Table).

## Physico-chemical characterizations of breeding habitats

Physico-chemical parameters (mean ± S.E) of different breeding habitat water bodies (pond, rice-fields & drain) in four seasons (summer, monsoon, post-monsoon & winter) are summarized in Table 2. Details of the data have been given in S1 File. During the whole study dissolve oxygen (D.O) content was found higher in pond and rice-field water than drain water, whereas, alkalinity and pH were found comparatively higher in more turbid drain water than less turbid pond and rice-field water bodies. Besides these drain water was found to be more hard containing more amount of nitrtate ($NO_3^-$) and phosphate ($PO_4^-$) ions than comparatively less hard pond and rice-field water. Drain water was found to contain more total dissolved solid (TDS) and had higher conductivity than rice-field and pond water (Table 2).

Results of one way ANOVA indicated that the physico-chemical parameters like dissolved oxygen (D.O), pH, alkalinity, total dissolved solids (tds), turbidity, hardness & conductivity and concentration of nitrate ($NO_3^-$) & phosphate ($PO_4^-$) showed significant differences between various habitat types viz. ponds, drains & rice-fields during summer (ANOVA, p<0.001), monsoon (ANOVA, p<0.001) & post-monsoon (ANOVA, p<0.001). Results of Mann-Whitney test also indicated that these physico-chemical parameters differ significantly between ponds and drains during winter season (Mann-Whitney test, p<0.001). Only the concentration of chloride ion did not show any significant variation between these habitats during summer, monsoon, post-monsoon (ANOVA, p>0.05) & winter (Mann-Whitney test, p>0.05). Temperature of habitat water also showed significant variations among different habitat types during summer (ANOVA, p = 0.0051), post-monsoon (ANOVA, p = 0.0011) & winter (Mann-Whitney test, p = 0.0029) seasons, but did not show any significant variation during monsoon season (ANOVA, p = 0.2536) (S2A–S2C to S5A–S5C Tables). Significant differences among these physico-chemical parameters between habitat types during summer, monsoon, post-monsoon and winter have been depicted in Figs 2–5.

Principle component analysis revealed the relationship between physico-chemical parameters of water and their associations with the larval density of *An. subpictus* mosquito (Fig 6A–6D). In case of summer the two major components of PCA (F1 and F2) together contributed 71.10% of the total variance (56.33% and 14.78% variation explained by F1 and F2 respectively) (Fig 6A). In this case the major factors that contributed F1 were turbidity (13.25%) followed by D.O (12.81%), nitrate (11.41%), alkalinity (11.08%) and phosphate (10.40%). The major factors responsible for construction of F2 were chloride (22.79%) followed by L.D (19.83%). Nitrate (0.87), alkalinity (0.86), phosphate (0.83), hardness (0.82), pH (0.81), TDS (0.75), and

**Table 2. Seasonwise physico-chemical parameters (Mean±S.E) of different habitat waterbodies of *Anopheles subpictus*.**

| Parameter | Summer | | | Monsoon | | | Post-monsoon | | | Winter | | |
|---|---|---|---|---|---|---|---|---|---|---|---|---|
| | Pond | Dain | Rice-field | Pond | Dain | Rice-field | Pond | Dain | Rice-field | Pond | Dain | Rice-field |
| Temperature (°C) | 31.29 ± 0.21 | 30.97 ± 0.17 | 30.3 ± 0.23 | 28.26 ± 0.19 | 28.49 ± 0.15 | 28.67 ± 0.16 | 27.79 ± 0.21 | 27.18 ± 0.21 | 28.17 ± 0.06 | 21.75 ± 0.23 | 20.74 ± 0.22 | - |
| D.O (mg/L) | 6.81 ± 0.13 | 4.44 ± 0.15 | 6.39 ± 0.08 | 7.79 ± 0.24 | 4.21 ± 0.14 | 7.34 ± 0.07 | 7.59 ± 0.14 | 4.57 ± 0.10 | 7.29 ± 0.08 | 7.38 ± 0.13 | 4.62 ± 0.15 | - |
| Alk (mg/L) | 133.5 ± 4.13 | 292.2 ± 5.50 | 183.9 ± 5 3.71 | 131.25 ± 9.25 | 239.95 ± 9.51 | 180.2 ± 2.78 | 125.9 ± 2.47 | 256.6 ± 55.84 | 228.7 ± 2.47 | 164.5 ± 4.43 | 260.9 ± 6.81 | - |
| pH | 7.01 ± 0.04 | 8.02 ± 0.03 | 7.54 ± 0.03 | 6.88 ± 0.08 | 7.95 ± 0.10 | 7.02 ± 0.02 | 6.96 ± 0.03 | 7.77 ± 0.05 | 7.25 ± 0.04 | 6.9 ± 0.08 | 7.85 ± 0.17 | - |
| Turb (NTU) | 3.58 ± 0.20 | 9.31 ± 0.41 | 4.21 ± 0.13 | 3.57 ± 0.29 | 10.16 ± 0.40 | 3.23 ± 0.06 | 3.83 ± 0.22 | 10.97 ± 0.41 | 4.04 ± 0.11 | 3.01 ± 0.15 | 9.52 ± 0.45 | - |
| TDS (mg/L) | 319.75 ± 8.53 | 474.4 ± 33.37 | 196.2 ± 3.79 | 350.75 ± 19.20 | 559.55 ± 22.48 | 404.8 ± 4.52 | 273.25 ± 17.41 | 413.35 ± 10.85 | 248.05 ± 6.77 | 229 ± 10.00 | 360 ± 17.45 | - |
| T.H (mg/L) | 235.8 ± 10.82 | 345.45 ± 19.07 | 212.45 ± 5.53 | 160.8 ± 10.34 | 411.85 ± 25.89 | 113.35 ± 2.38 | 234.15 ± 10.58 | 439.45 ± 22.83 | 133.95 ± 3.17 | 206.35 ± 11.22 | 395.7 ± 22.58 | - |
| E.C (µs/cm) | 272.1 ± 11.53 | 347.6 ± 24.59 | 278.05 ± 12.95 | 279.1 ± 12.27 | 391.5 ± 22.28 | 185.7 ± 5.36 | 332.15 ± 13.32 | 507.6 ± 25.02 | 375.9 ± 5.40 | 232 ± 11.86 | 313.25 ± 14.46 | - |
| Cl⁻ (ppm) | 38.01 ± 2.70 | 40.22 ± 2.26 | 34.38 ± 1.00 | 41.22 ± 2.21 | 38.34 ± 1.30 | 40.83 ± 1.33 | 47.90 ± 2.85 | 49.84 ± 3.31 | 47.91 ± 1.55 | 40.91 ± 3.00 | 41.44 ± 2.57 | - |
| NO₃⁻ (ppm) | 0.06 ± 0.007 | 2.88 ± 0.36 | 0.15 ± 0.03 | 0.22 ± 0.03 | 2.82 ± 0.36 | 0.42 ± 0.04 | 0.34 ± 0.05 | 3.86 ± 0.29 | 0.95 ± 0.05 | 0.05 ± 0.007 | 2.13 ± 0.27 | - |
| PO₄⁻ (ppm) | 0.24 ± 0.03 | 4.02 ± 0.48 | 0.73 ± 0.07 | 0.26 ± 0.03 | 2.49 ± 0.35 | 0.68 ± 0.06 | 1.36 ± 0.16 | 3.54 ± 0.32 | 1.14 ± 0.15 | 0.18 ± 0.02 | 2.33 ± 0.28 | - |

The values are average (mean± S.E) of twenty habitats.

* Due to absence of water in the rice-fields of study areas, it could not be transformed into larval habitat during winter season.

Where, temp = water temperature, Alk = alkalinity, D.O = dissolved oxygen, E.C. = electrical conductivity, T.H = total hardness, TDS = total dissolved solids, Turb = turbidity, Cl⁻ = chloride content, PO₄⁻ = phosphate content, NO₃⁻ = Nitrate content.

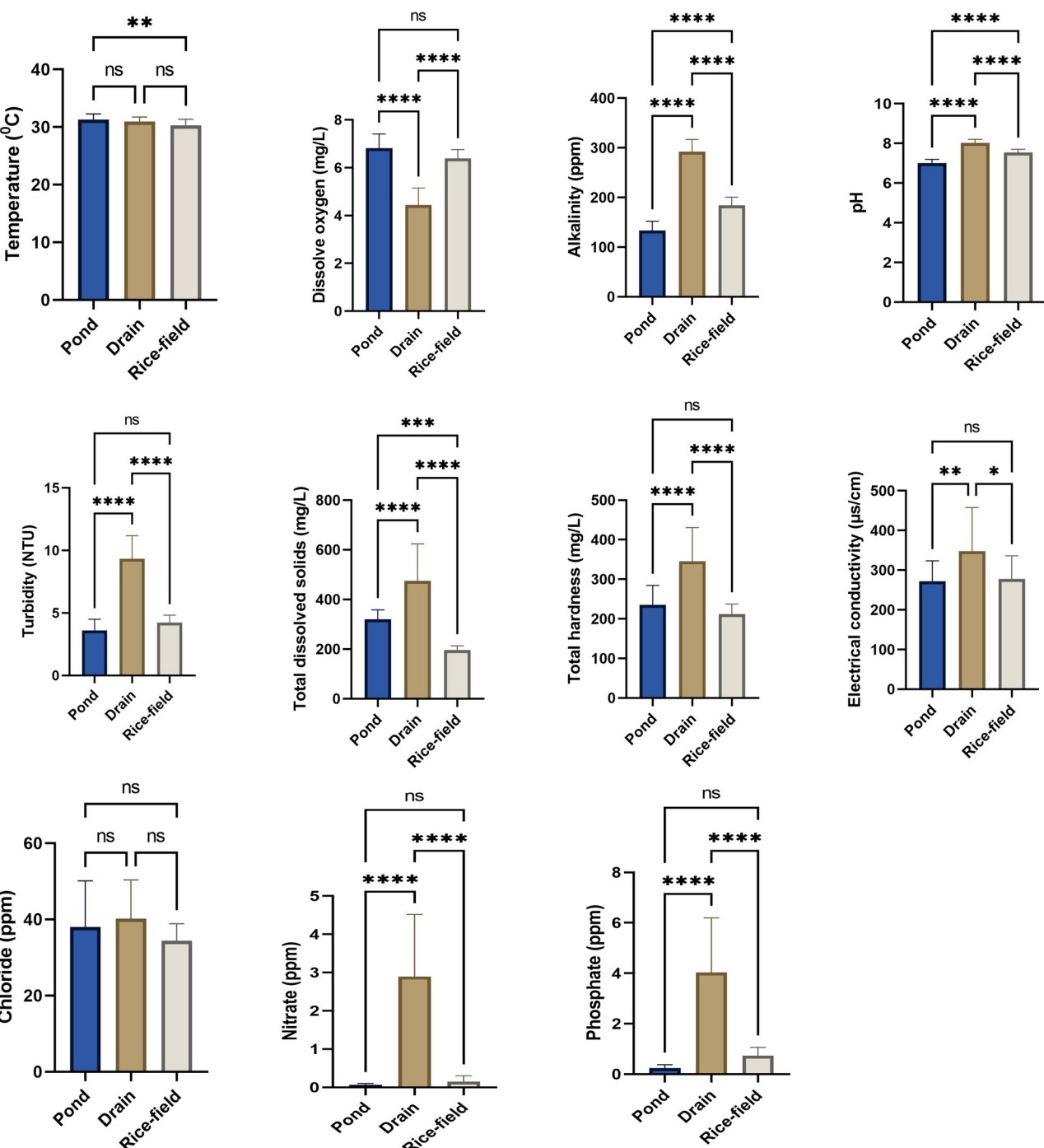

**Fig 2. Physico-chemical parameters of different habitat types during summer season.** An asterisk above given bar indicates significant difference. '****' indicates p<0.0001, '***' indicates p<0.001, '**' indicates p< 0.01, 'ns' indicates p>0.05.

conductivity (0.60) were found to positively corelated with F1, whereas, D.O (-0.93) & L.D (-0.65) were found to be negatively correlated with F1. On the other hand chloride (0.63), L.D (0.59) & TDS (0.52) were found to be positively correlated with F2, while pH (-0.48) & alkalinity (-0.34) were negatively corelated with F2 (S6A–S6C Table). During monsoon season F1

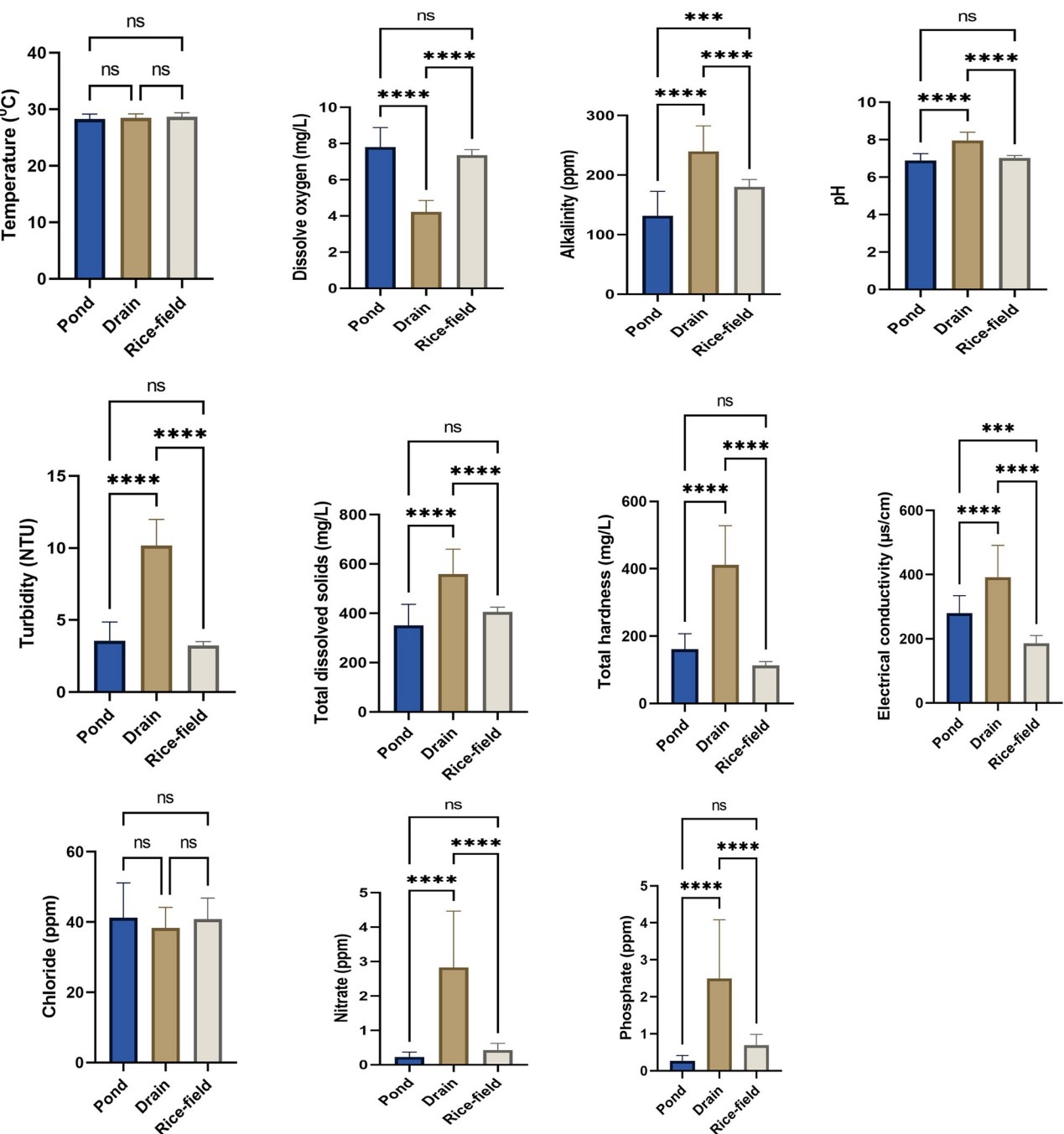

**Fig 3. Physico-chemical parameters of different habitat types during monsoon season.** An asterisk above given bar indicates significant difference. '****' indicates p<0.0001, '***' indicates p<0.001, '*' indicates p< 0.01, 'ns' indicates p>0.05.

and F2 together contributed 72.55% of the total variance (60.53% and 12.02% variation explained by F1 and F2 respectively) (Fig 6B). Here, the major contributing factors of F1 were turbidity (12.48%) followed by hardness (12.18%), D.O (11.94%), nitrate (10.54%), pH (10.48%) and phosphate (10.35%), whereas factors that contributed in F2 mostly were temperature (34.30%) & L.D (31.94%). In this plot, parameters that showed positive association with F1 were turbidity (0.95), hardness (0.94), nitrate (0.87), pH (0.87), phosphate (0.86) &

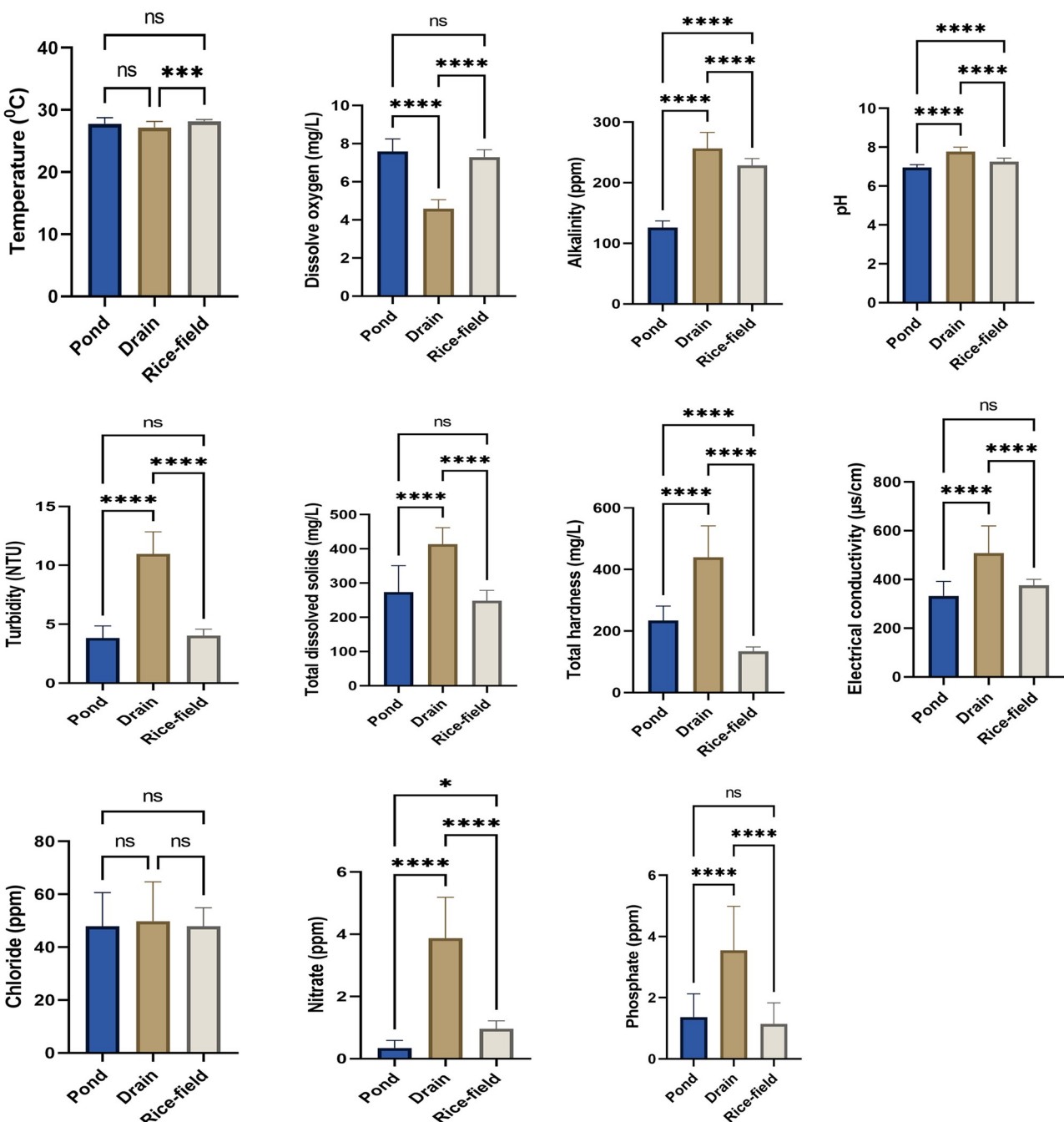

**Fig 4. Physico-chemical parameters of different habitat types during post-monsoon season.** An asterisk above given bar indicates significant difference. '****' indicates $p < 0.0001$, '***' indicates $p < 0.001$, '**' indicates $p < 0.01$, 'ns' indicates $p > 0.05$.

alkalinity (0.84), whereas negative association of F1 was found with D.O (-0.93) & L.D (-0.55). F2 showed positive association with temperature (0.70) & alkalinity (0.36) and negative association with L.D (-0.67) & conductivity (-0.43) (S7A–S7C Table). During post-monsoon season F1 and F2 together contributed 73.83% of the total variance (60.59% and 13.24% variation explained by F1 and F2 respectively) (Fig 6C). Here major contributing factors for F1 were turbidity (12.24%), nitrate (11.90%), D.O (11.63%) & hardness (10.07%). The major factors

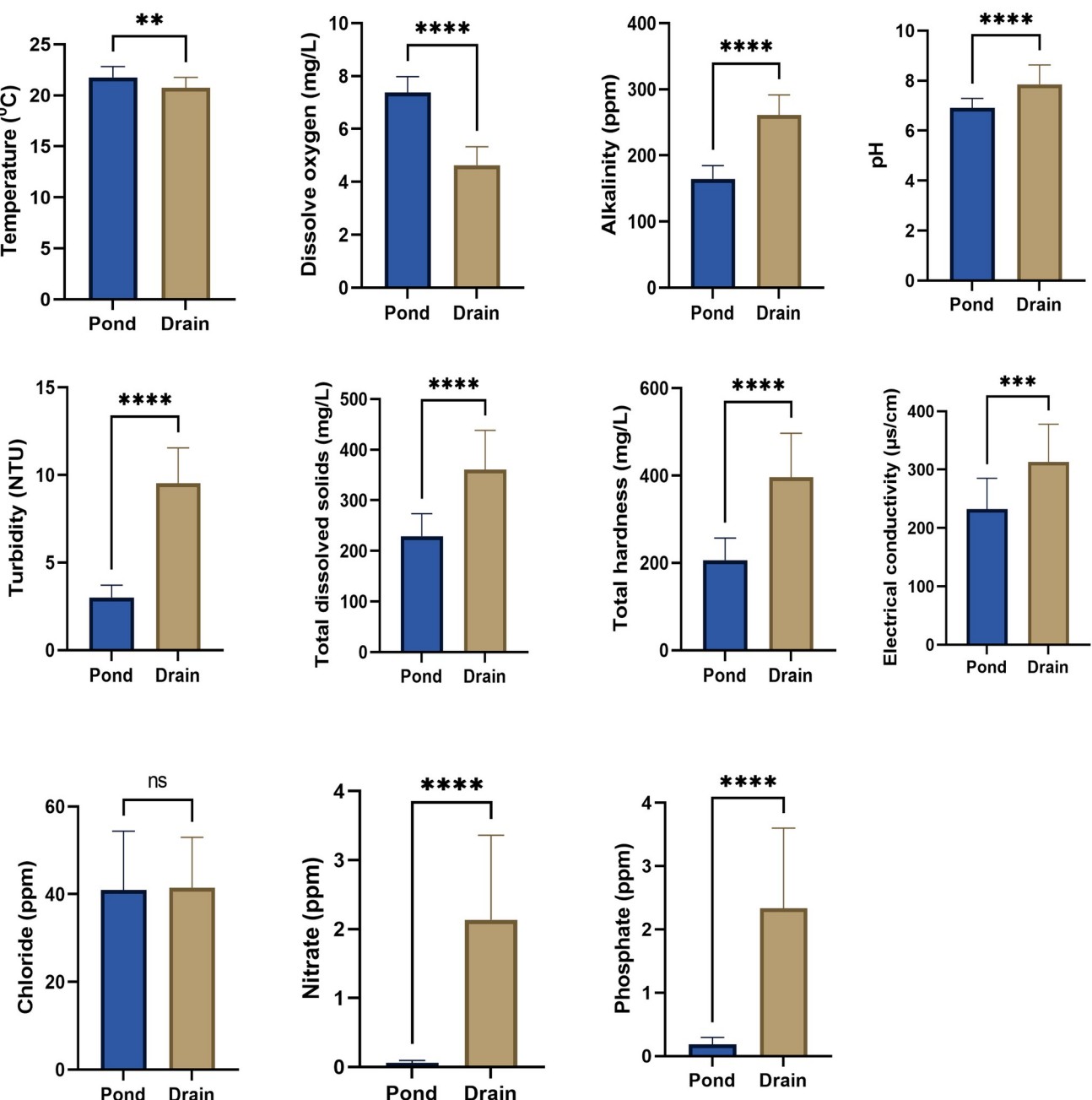

**Fig 5. Physico-chemical parameters of different habitat types during winter season.** An asterisk above given bar indicates significant difference. '****' indicates p<0.0001, '***' indicates p<0.001, '**' indicates p< 0.01, 'ns' indicates p>0.05.

contributing F2 were L.D (29.82%) & alkalinity (23.52%). Parameters that showed positive correlation with F1 were turbidity (0.94), nitrate (0.93), hardness (0.85), conductivity (0.84), phosphate (0.83) & TDS (0.82), whereas, negative correlation with F1 were shown by D.O (-0.92) & LD (-0.62). On the other hand L.D (0.68) & alkalinity (-0.61) showed positive & negative correlation respectively with F2 (S8A–S8C Table). During winter season the F1 and F2 together showed 71.73% of the total variance (60.96% and 10.77% variation explained by F1 and F2 respectively) (Fig 6D). In this case, alkalinity contributed most (12.04%) for F1 followed by D.

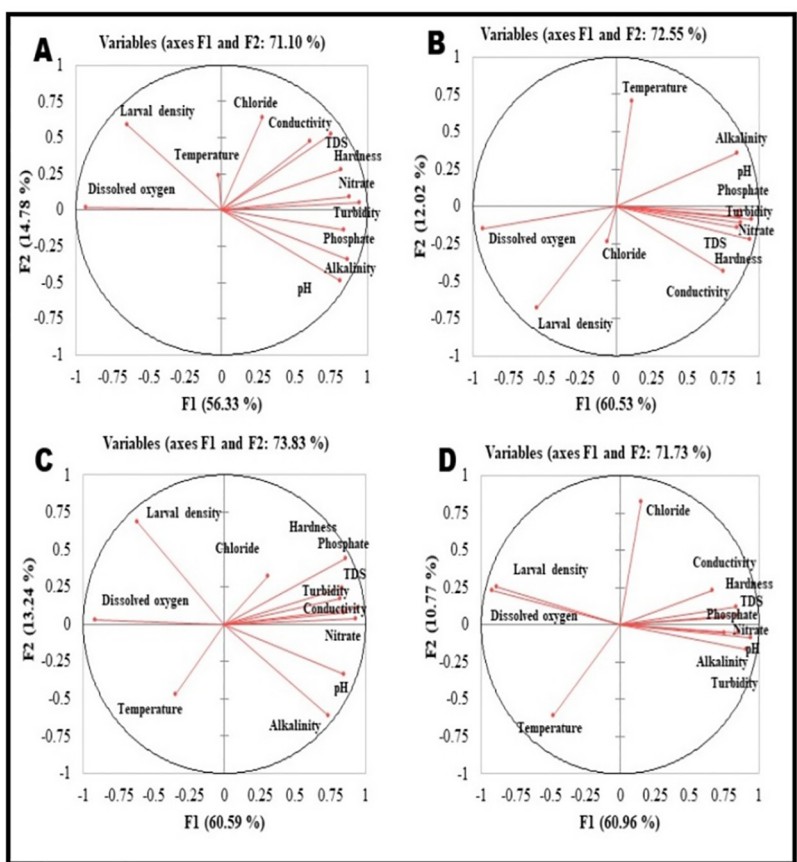

**Fig 6. Principal component analysis showing the relationship between the physicochemical parameters of water and the larval density of *Anopheles subpictus* (A = summer, B = monsoon, C = post-monsoon, D = winter).**

O (11.66%), turbidity (11.24%) & L.D (10.87%), whereas for F2 two major contributing factors were chloride (52.89%) & temperature (29.03%). Here positive correlation with F1 were shown by alkalinity (0.93), turbidity (0.90), phosphate (0.85), nitrate (0.84) & hardness (0.83), whereas, D.O (-0.92) & L.D (-0.89) showed negative correlation with F1. On the other hand, parameters like chloride (0.82) & temperature (-0.61) showed positive & negative correlation respectively with F2 (S9A–S9C Table). In all the four seasons larval density (L.D) showed positive association with the dissolved oxygen (D.O) content of water. Larval density of all four seasons showed orthogonal location with chloride content (Cl⁻) of habitat water which indicated no significant correlation between them. Temperature variation of habitat water, irrespective of habitat types did not show any significant influence on larval density in a particular season. Rest of the physico-chemical parameters such as alkalinity, pH, turbidity, total dissolved solids (TDS), total hardness, electrical conductivity, nitrate ($NO_3^{3-}$) & phosphate ($PO^{4-}$) content showed opposite direction of larval density, indicated negative correlation of these parameters with the larval density (Fig 6A–6D).

Pearson correlation indicated significant correlation of larval density of *An. subpictus* with physico-chemical parameters of habitat water in different seasons. Larval density of *An. subpictus* showed significant positive correlation with dissolved oxygen content of habitat water and significant negative correlation with alkalinity of habitat water in all the four seasons (Table 3).

**Table 3. Correlation between larval density of *Anopheles subpictus* and physicochemical parameters of habitat water.**

| Parameter | Summer | | Monsoon | | Post-monsoon | | Winter | |
|---|---|---|---|---|---|---|---|---|
| | r value | p value | r value | p value | r value | p value | r value | p value |
| Temperature | 0.27236 | 1 | -0.30865 | 1 | -0.04533 | 1 | 0.29563 | 1 |
| pH | **-0.82034** | 6.99E-14* | -0.44403 | 0.028182 | **-0.63416** | 3.52E-06* | **-0.57601** | 0.006636* |
| Alkalinity | **-0.71716** | 7.58E-09* | **-0.72865** | 4.01E-09* | **-0.87316** | 6.22E-18* | **-0.86135** | 6.50E-11* |
| D.O | **0.63089** | 4.32E-06* | **0.58082** | 9.35E-05* | **0.60058** | 2.59E-05* | **0.93816** | 2.82E-17* |
| Conductivity | -0.26941 | 1 | -0.066031 | 1 | **-0.5321** | 0.00079961* | **-0.50269** | 0.062527* |
| Hardness | -0.323 | 0.78081 | -0.31125 | 1 | -0.20706 | 1 | **-0.73508** | 4.36E-06* |
| TDS | -0.10393 | 1 | -0.44958 | 0.023368 | -0.31199 | 1 | **-0.59929** | 0.0028963* |
| Turbidity | **-0.56463** | 0.00017244* | -0.43598 | 0.03678 | -0.47284 | 0.0089892* | **-0.87773** | 6.92E-12* |
| Chloride | 0.013432 | 1 | 0.12051 | 1 | -0.10912 | 1 | 0.02208 | 1 |
| Phosphate | **-0.51092** | 0.0019996* | -0.42093 | 0.059474 | -0.34836 | 0.42097 | **-0.69932** | 3.41E-05* |
| Nitrate | -0.45911 | 0.014821 | -0.38644 | 0.16509 | **-0.56364** | 0.00018109* | **-0.70556** | 2.43E-05* |

The values given here are Pearson's correlation (r) and significance level (p).

Values (r) in bold and asterisk (*) above a given value (p) indicates significant correlation at p = <0.01 after Holms-Bonferroni correction.

The results of GLM analysis showed that there was a significant interaction of larval density (L.D) of *An. subpictus* with pH (F = 5.60, p = 0.01), D.O (F = 26.70, p = 0.000001) & alkalinity (F = 72.47, p = 0.000000) of habitat water (S10A–S10C Table and Fig 7). The adjusted $R^2$ value for selected model is 0.59 (F = 29.93, p = 0.001). Prediction equation from this model is L.D = 19.5339256726 + 0.130674497035 × "Temperature" + 2.05506521603 × pH -0.0788381332804 × "Alkalinity" + 2.32091156053 × "D.O" + 0.00151644594031 × "Conductivity" + 0.00622796606098 × Hardness + 0.00326658747068 × TDS + 0.448792969762 × "Turbidity" + 0.0133691992506 × Chloride + 0.377903805823 × "Phosphate" -0.0483529968281 × Nitrate (S10 Table).

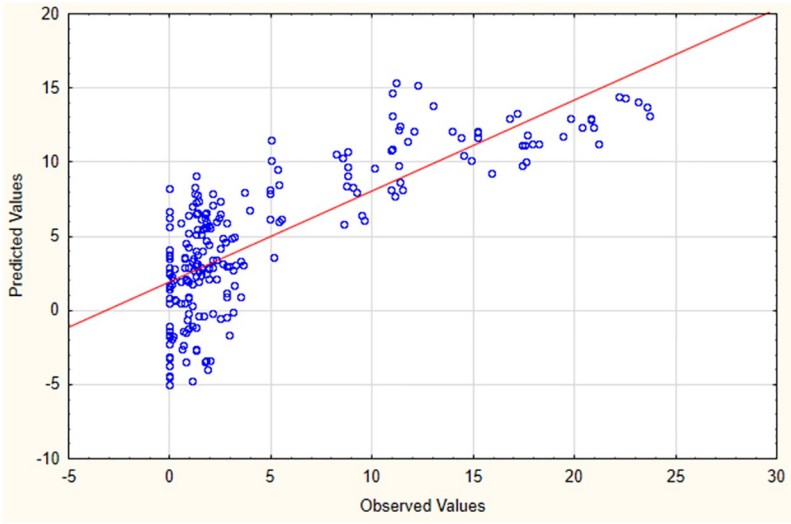

**Fig 7. Observed vs. predicted values of larval density according to Generalized Linear Model (GLM) (Multiple $R^2$ value 0.61).**

## Populations of different bacterial groups in breeding habitats

Different groups of bacterial populations ($\times 10^3$ cfu/mL of water) viz, aerobic heterotrophic, *Bacillus*, Gram negative, protein hydrolyzing, starch hydrolyzing & nitrate reducing bacteria in different types of habitat water viz., pond, drain & rice-field in four different seasons have been depicted in Table 4 (S1 Fig). Details of data have been listed in S2 File. Pond water that exhibited higher larval density of *An. subpictus*, were also recorded to have comparatively higher populations of *Bacillus* group of bacteria in all the four seasons during the study period.

## Bacterial characterizations of breeding habitats

During one year study period, microbiological examinations of different breeding habitats of anopheline larvae revealed altogether twenty-five bacterial isolates and they were named as HABW1, HABW2, HABW3, HABW4, HABW5, HABW6, HABW7, HABW8, HABW9, HABW10, HABW11, HABW12, HABW13, HABW14, HABW15, HABW16, HABW17, HABW18, HABW19, HABW20, HABW21, HABW22, HABW23, HABW24 and HABW25. Out of which eight bacterial isolates (HABW1, HABW3, HABW4, HABW6, HABW10, HABW12, HABW14 and HABW15) were screened from all the breeding habitat water irrespective of habitat type in the presence of anopheline larvae throughout the year. Colony morphologies of these bacterial isolates have been provided in Table 5.

**Table 4. Season wise different groups of bacterial populations in the form of colony forming unit (cfu) in the breeding habitats of *Anopheles subpictus* mosquitoes.**

| Season | Habitat type | A.H | *Bacillus* | G.N | P.H | S.H | N.R |
|---|---|---|---|---|---|---|---|
| | | (Mean ±S.E) | (Mean± S. E) | (Mean± S.E) | (Mean± S.E) | (Mean± S.E) | (Mean± S.E) |
| **SUMMER** | POND WATER (cfu/mL×10^3) | 52.58 ±1.15 | 32.05 ±1.25 | 1.48 ±0.09 | 0.94± 0.05 | 5.36 ±0.20 | 3.39± 0.19 |
| | RICE-FIELD WATER (cfu/mL×10^3) | 48.75 ±0.67 | 27.01 ±0.49 | 1.15 ±0.12 | 0.98 ±0.05 | 2.74± 0.07 | 1.57 ±0.14 |
| | DRAIN WATER (cfu/mL×10^3) | 58.56 ±1.90 | 24.55 ±1.19 | 2.73 ±0.21 | 1.01 ±0.05 | 4.85 ±0.12 | 2.77 ±0.16 |
| **MONSOON** | POND WATER (cfu/mL×10^3) | 56.61 ±2.46 | 29.26 ±2.48 | 2.69± 0.11 | 1.06± 0.07 | 8.18 ±0.46 | 3.41± 0.15 |
| | RICE-FIELD WATER (cfu/mL×10^3) | 46.67 ±0.87 | 23.26 ±0.95 | 1.90 ±0.18 | 0.87± 0.06 | 5.42 ±0.19 | 1.55 ±0.10 |
| | DRAIN WATER (cfu/mL×10^3) | 60.76 ±2.14 | 23.16 ±1.70 | 4.68± 0.24 | 0.85± 0.04 | 4.89 ±0.27 | 2.05± 0.15 |
| **POST-MONSOON** | POND WATER (cfu/mL×10^3) | 40.88 ±2.57 | 16.29 ±0.99 | 1.31 ±0.09 | 0.75± 0.04 | 4.33 ±0.27 | 2.36 ±0.21 |
| | RICE-FIELD WATER (cfu/mL×10^3) | 33.12± 2.10 | 8.35±0.29 | 0.80± 0.06 | 0.79± 0.04 | 3.38± 0.19 | 0.87 ±0.04 |
| | DRAIN WATER (cfu/mL×10^3) | 51.92± 2.68 | 10.39 ±0.44 | 2.34 ±0.19 | 0.86± 0.05 | 4.84 ±0.19 | 2.72± 0.17 |
| **WINTER** | POND WATER (cfu/mL×10^3) | 22.47± 0.86 | 7.69±0.60 | 0.85 ±0.04 | 0.58 ±0.04 | 3.39± 0.15 | 1.15± 0.05 |
| | DRAIN WATER (cfu/mL×10^3) | 32.92± 1.86 | 5.16±0.34 | 1.01 ±0.03 | 0.68 ±0.04 | 2.30± 0.12 | 1.12 ±0.05 |
| | RICE-FIELD WATER (cfu/mL×10^3) | - | - | - | - | - | - |

The values are averages (mean ± S.E) of twenty replications.

A.H.- Aerobic Heterotrophic bacteria, G.N.-Gram Negative bacteria, P.H.- Protein Hydrolyzing bacteria, S.H.- Starch Hydrolyzing bacteria, N.R.- Nitrate Reducing bacteria.

**Table 5. Colony characteristics of common bacterial isolates from breeding habitats of *Anopheles subpictus* mosquitoes.**

| SI No. | Name of isolates | Colony Characters | | | | | | |
|---|---|---|---|---|---|---|---|---|
| | | Shape | Size (mm) (Mean±S.D) | Opacity | Elevation | Consistency | Margin | Colour |
| 1 | HABW1 | Irregular | 3.52±0.35 | Opaque | Slightly elevated | Buttery | Wavy | Off white |
| 2 | HABW3 | Round | 2.56±0.32 | Opaque | Flat | Mucoid | Irregular | Greenish white |
| 3 | HABW4 | Round | 2.46±0.27 | Opaque | Slightly elevated | Mucoid | Smooth | Yellowish White |
| 4 | HABW6 | Round | 2.04±0.13 | Opaque | Elevated | Moist | Smooth | Creamy white |
| 5 | HABW10 | Irregular | 2.62±0.23 | Opaque | Flat | Dry | Wrinkled | White |
| 6 | HABW12 | Round | 2.12±0.28 | Opaque | Slightly elevated | Moist | Smooth | Greyish white |
| 7 | HABW14 | Round | 2.34±0.20 | Opaque | Flat | Dry | Undulate | White |
| 8 | HABW15 | Round | 1.62± 0.23 | Opaque | Elevated | Mucoid | Smooth | Pale yellow |

## Staining properties of bacterial isolates

Among these eight bacterial isolates, four isolates were found to be Gram positive and able to produce endospores, whereas rest of the four isolates were Gram negative and not able to produce endospores (Table 6 and S2 Fig).

## Bio-chemical characterization of bacterial isolates

The results of the biochemical tests of the eight resident bacterial strains are summarized in Table 7.

## Ovipositional bioassay

Ovipositional response of gravid *An. subpictus* mosquitoes towards eight common breeding habitat bacterial isolates were evaluated in laboratory condition by dual choice egg count bioassay method. Out of the eight isolates, *An. subpictus* showed significant positive oviposition response towards test cup containing suspension of HABW1 ($t_{(17.92, 4)}$, p<0.0001), HABW4 ($t_{(9.664,4)} = 0.0006$), HABW10 ($t_{(23.26, 4)}$, p<0.0001) & HABW14 ($t_{(5.786, 4)}$, p = 0.0044)

**Table 6. Staining properties of common bacterial isolates from breeding habitats of *Anopheles subpictus* mosquito.**

| Name of the Isolates | Gram's Stain | | Endospore |
|---|---|---|---|
| | Property | Shape of Vegetative cells | |
| HABW1 | Positive | Rods either single or in short chain | Present |
| HABW3 | Negative | Rods single | Absent |
| HABW4 | Positive | Rods in long chain | Present |
| HABW6 | Negative | Rods single | Absent |
| HABW10 | Positive | Rods single or in short chain | Present |
| HABW12 | Negative | Rods single | Absent |
| HABW14 | Positive | Rods single | Present |
| HABW15 | Negative | Cocco-bacilli | Absent |

**Table 7. Biochemical properties of common bacterial isolates from breeding habitats of *Anopheles subpictus* mosquito.**

| Name of the bio-chemical tests | Name of the bacterial isolates | | | | | | | |
|---|---|---|---|---|---|---|---|---|
| | HABW1 | HABW3 | HABW4 | HABW6 | HABW10 | HABW12 | HABW14 | HABW15 |
| Catalase production | + | + | + | + | + | + | + | + |
| Indole production | - | - | - | - | - | + | - | - |
| Methyl Red (MR) | + | - | + | - | - | + | - | - |
| Voges Proskauer (VP) | - | - | - | + | + | - | + | - |
| Citrate utilization | - | + | - | + | + | - | + | + |
| Nitrate reduction | + | + | - | + | + | + | + | - |
| Oxidase production | + | + | + | - | + | - | + | - |
| Urease production | - | - | + | - | - | - | - | - |
| Starch hydrolysis | + | - | - | + | + | - | + | - |
| Lipid hydrolysis | + | + | - | - | + | - | + | - |
| Gelatine hydrolysis | + | + | + | - | + | - | + | - |
| Motility | + | + | + | + | + | + | + | - |
| Triple Sugar Iron Agar (TSI) | K/A | K/K | K/A | A/A, G | A/A | A/A, G | A/A | K/A |
| $H_2S$ production | - | - | - | - | - | - | - | - |

+ = Positive, − = Negative, K = Alkaline, A = Acidic, G = Gas.

(Table 8). Number of eggs laid in test cups containing other four bacterial isolates (HABW3, HABW6, HABW12 & HABW15) did not show any significant difference when compared with the number of eggs laid in their respective control cups (paired t-test, p>0.05) (Table 8). The oviposition activity index was 0.79, 0.62, 0.80 & 0.62 in case of HABW1, HABW4, HABW10 & HABW14 respectively. These values were 0.52, 0.51, 0.51 & 0.48 in case of HABW3, HABW6, HABW12 & HABW15 respectively. Further, when the mosquitoes were provided with a mixed suspension of all eight bacterial isolates, they also laid significant higher number of eggs in test cups than control cups (paired t-test, p<0.0001) and the oviposition activity index was 0.81 (Table 8). Details of data in relation to mosquito oviposition have been given in S3 File.

**Table 8. Oviposition response of *Anopheles subpictus* to bacterial isolates in dual choice bioassay.**

| Bacterial isolates | No. of eggs laid (Mean±S.E) | | Oviposition Activity Index (OAI) | p value* |
|---|---|---|---|---|
| | Control cups | Test Cups | | |
| **HABW1** | 62.6±9.45 | 245.8±18.58 | 0.79 | **<0.0001** |
| **HABW3** | 142.6±8.34 | 157.6±11.32 | 0.52 | 0.4614 |
| **HABW4** | 129±9.45 | 215±11.31 | 0.62 | **0.0006** |
| **HABW6** | 91.2±6.73 | 96.2±7.33 | 0.51 | 0.4527 |
| **HABW10** | 90±7.30 | 377±10.27 | 0.80 | **<0.0001** |
| **HABW12** | 142.6±11.33 | 149.8±12.63 | 0.51 | 0.7641 |
| **HABW14** | 106.4±7.97 | 176.8±6.63 | 0.62 | **0.0044** |
| **HABW15** | 131.6±12.91 | 123.4±8.36 | 0.48 | 0.6255 |
| **All Isolates** | 89.6±8.32 | 392.2±14.67 | 0.81 | **<0.0001** |

*The mean numbers of eggs laid in control cups and test cups are shown by paired t- test at p<0.05.

## Molecular characterizations of bacterial isolates

Among the eight bacterial isolates which were found to be common in all habitat types of *Anopheles subpictus* throughout the year, four bacterial isolates (HABW1, HABW4, HABW10 & HABW14) were recorded as potent oviposition attractant of *Anopheles subpictus* mosquito. Therefore, these four bacterial isolates were further characterized by molecular methods to confirm their identification. The obtained 16S rDNA nucleotide sequences of these bacterial isolates were submitted to NCBI GenBank database and the following accession numbers have been allotted: MN153450 MN173350 MN153430 & MZ363639 to HABW1, HABW4, HABW10 & HABW14, respectively.

Phylogenetic analysis through neighbour-joining method indicated that *Bacillus cereus* HABW1 (MN153450) closely related to *B. cereus* (MH210863), whereas, ML method indicated that the bacterial isolate *B. cereus* HABW1 (MN153450) was closely similar to *B. cereus* (HQ684015) (Fig 8). Neighbour-joining tree of *Bacillus megaterium* HABW4 (MN173350) depicted that this bacteria is closely related with *B. megaterium* (KX495254) and according to ML tree this bacterial isolate is closely related with *B. megaterium* (KP017584) & *B. megaterium* (HQ634276) (Fig 9). Both neighbour-joining and ML tree of *Bacillus subtilis* HABW10 (MN166905) indicated that this bacterial strain is closely related to *B. subtilis* (EF633176) (Fig 10). Phylogenetic tree prepared by both neighbour-joining and ML method indicated that *Bacillus tequilensis* HABW14 (MZ363639) closely related with *B. tequilensis* (MK018119) (Fig 11).

## Scanning electron microscopic analysis of bacterial isolates

Scanning electron microscopic images revealed that all the organisms of the four bacterial isolates were rod shaped either single or in small or long chain (Fig 12) and they produced round or oval shaped endospores (Fig 13).

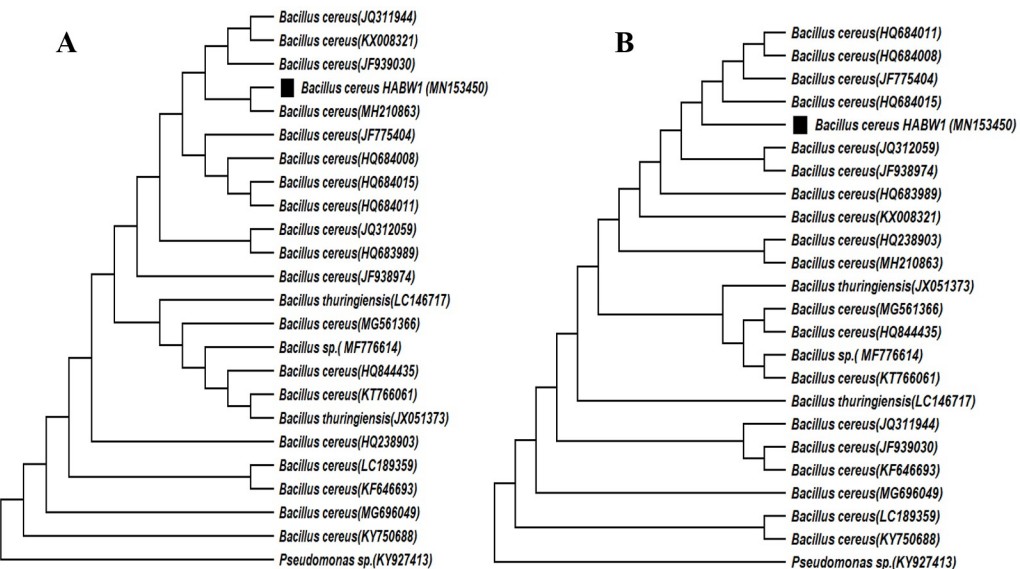

**Fig 8. Phylogenetic tree (A: Neighbour-joining method, B: Maximum likelihood method) constructed based on 16S rRNA gene sequences of *Bacillus cereus* HABW1 (MN153450) and other related sequences retrieved from NCBI.**

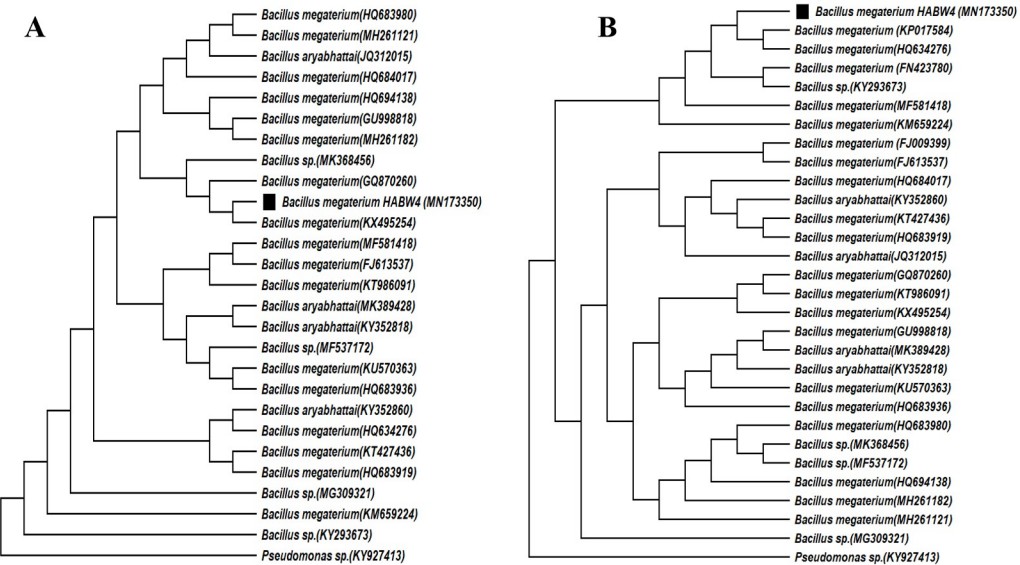

**Fig 9. Phylogenetic tree (A: Neighbour-joining method, B: Maximum likelihood method) constructed based on 16S rRNA gene sequences of *Bacillus megaterium* HABW4 (MN173350) and other related sequences retrieved from NCBI.**

## Carbohydrate fermentation ability of bacterial isolates

Fermentation of twenty different carbohydrate sources indicated that, all of the four bacterial isolates were able to ferment dextrose, fructose & trehalose but none of them could ferment dulcitol, adonitol, galactose, & rhamnose (Table 9).

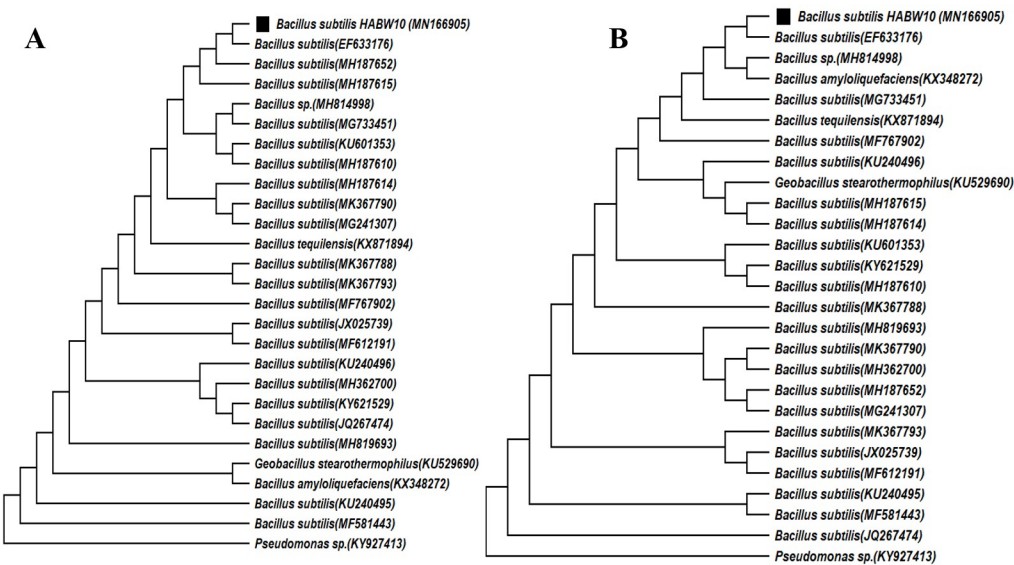

**Fig 10. Phylogenetic tree (A: Neighbour-joining method, B: Maximum likelihood method) constructed based on 16S rRNA gene sequences of *Bacillus subtilis* HABW10 (MN166905) and other related sequences retrieved from NCBI.**

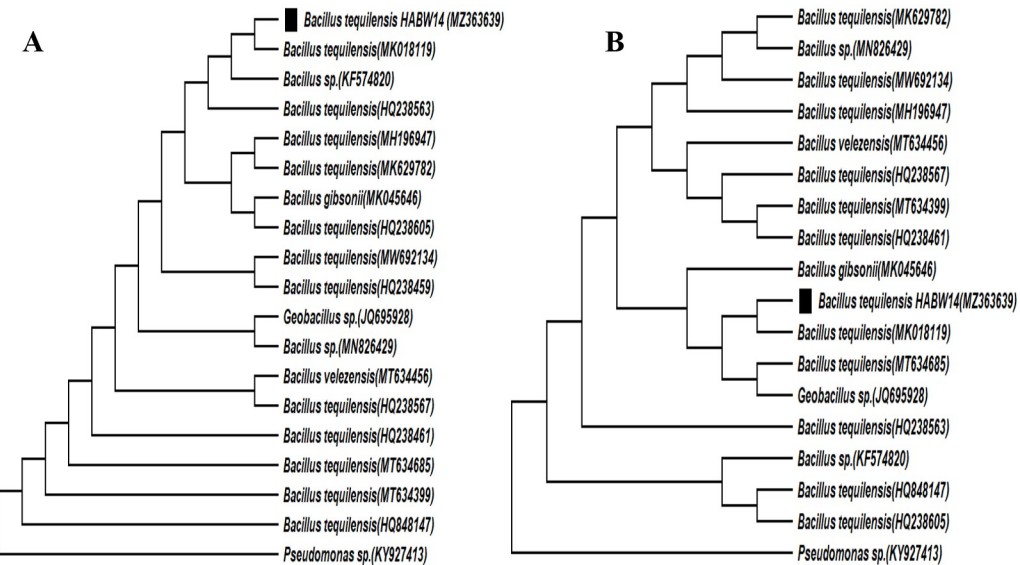

**Fig 11. Phylogenetic tree (A: Neighbour-joining method, B: Maximum likelihood method) constructed based on 16S rRNA gene sequences of *Bacillus tequilensis* HABW14 (MZ363639) and other related sequences retrieved from NCBI.**

## Antibiotic sensitivity of bacterial isolates

All the oviposition attractant bacterial isolates of *Anopheles subpictus* i.e. *Bacillus cereus* HABW1 (MN153450), *Bacillus megaterium* HABW4 (MN173350), *Bacillus subtilis* HABW10 (MN166905) & *Bacillus tequilensis* HABW14 (MZ363639) were sensitive to standard dose of bacitracin (10 μg/disc), azithromycin (30 μg/disc), ciprofloxacin (5 μg/disc), chloramphenicol (30 μg/disc), doxycycline (30 μg/disc), gentamicin (50 μg/disc), erythromycin (15 μg/disc),

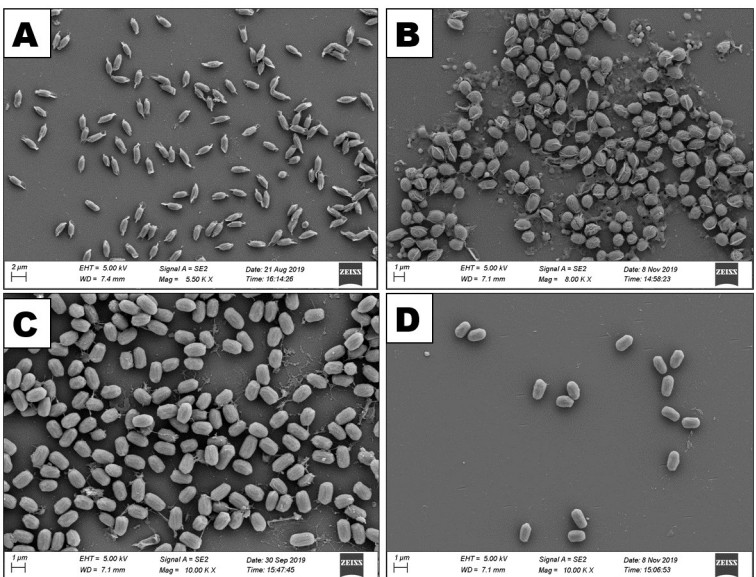

**Fig 12. Scanning electron micrograph showing vegetative bodies of the bacterial isolates.** A: *Bacillus cereus* HABW1, B: *Bacillus megaterium* HABW4, C: *Bacillus subtilis* HABW10, D: *Bacillus tequilensis* HABW14.

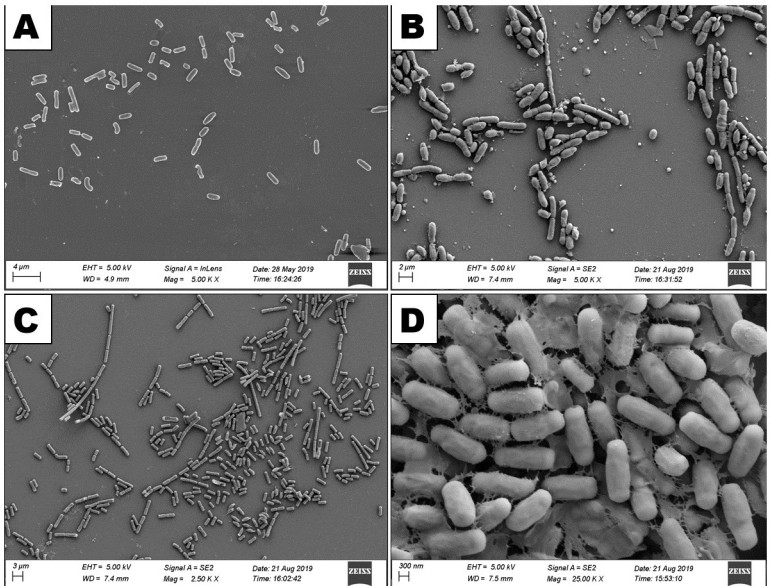

**Fig 13. Scanning electron micrograph showing endospores of the bacterial isolates.** A: *Bacillus cereus* HABW1, B: *Bacillus megaterium* HABW4, C: *Bacillus subtilis* HABW10, D: *Bacillus tequilensis* HABW14.

Table 9. Carbohydrate fermentation test of oviposition attractant bacterial isolates.

| Carbohydrate source | *Bacillus cereus* | *Bacillus megaterium* | *Bacillus subtilis* | *Bacillus tequilensis* |
|---|---|---|---|---|
| | HABW1 | HABW4 | HABW10 | HABW14 |
| **Dextrose (De)** | + | + | + | + |
| **Sucrose (Su)** | - | + | + | + |
| **Lactose (La)** | - | + | - | - |
| **Fructose (Fc)** | + | + | + | + |
| **Galactose (Ga)** | - | - | - | - |
| **Inositol (IS)** | - | - | + | + |
| **Mannitol (Mn)** | - | + | + | + |
| **Arabinose (Ar)** | - | - | + | - |
| **Cellobiose (Ce)** | - | - | + | + |
| **Trehalose (Te)** | + | + | + | + |
| **Dulcitol (Du)** | - | - | - | - |
| **Raffinose (Rf)** | - | + | + | - |
| **Melibiose (Mb)** | - | + | - | - |
| **Sorbitol (Sb)** | - | - | + | + |
| **Xylose (Xy)** | - | + | - | - |
| **Rhamnose (Rh)** | - | - | - | - |
| **Adonitol (Ad)** | - | - | - | - |
| **Mannose (Mo)** | - | + | + | + |
| **Salicin (Sa)** | - | + | - | - |
| **Inulin (In)** | - | + | + | + |

+ = positive, − = negative.

**Table 10. Antibiotic sensitivity of oviposition attractant bacteria of *Anopheles subpictus*.**

| Antibiotics | *B. cereus* | | *B. megaterium* | | *B. subtilis* | | *B. tequilensis* | |
|---|---|---|---|---|---|---|---|---|
| | HABW1 | | HABW4 | | HABW10 | | HABW14 | |
| | Sensitivity | ZDI (mm) | Sensitivity | ZDI (mm) | Sensitivity | ZDI (mm) | Sensitivity | ZDI (mm) |
| Chloramphenicol (C,30) | S | 18 | S | 22 | S | 28 | S | 30 |
| Kanamycin (K,30) | S | 16 | S | 20 | S | 22 | S | 26 |
| Levofloxacin (LE,5) | S | 30 | S | 22 | S | 38 | S | 37 |
| Gentamicin (GEN,50) | S | 25 | S | 22 | S | 25 | S | 25 |
| Neomycin (N,30) | S | 21 | S | 19 | S | 22 | S | 21 |
| Bacitracin (B, 10) | S | 9 | S | 17 | S | 10 | S | 13 |
| Ofloxacin (OF,5) | S | 28 | S | 18 | S | 33 | S | 33 |
| Norfloxacin (NX,10) | S | 29 | S | 17 | S | 33 | S | 33 |
| Tetracycline (TE,30) | S | 24 | S | 24 | S | 27 | S | 29 |
| Ciprofloxacin (CIP, 5) | S | 32 | S | 22 | S | 38 | S | 36 |
| Vancomycin (VA, 30) | S | 15 | S | 19 | S | 21 | S | 23 |
| Rifampicin (RIF,5) | S | 11 | S | 17 | S | 29 | S | 19 |
| Azithromycin (AZM,30) | S | 18 | S | 24 | S | 29 | S | 28 |
| Erythromycin (E,15) | S | 15 | S | 21 | S | 23 | S | 26 |
| Amoxicillin (AMX,10) | R | Nil | S | 13 | S | 30 | S | 18 |
| Ampicillin (AMP,10) | R | Nil | R | Nil | R | Nil | R | Nil |
| Penicillin (P,10) | R | Nil | R | Nil | R | Nil | R | Nil |
| Streptomycin (S,10) | S | 23 | S | 18 | S | 18 | S | 20 |
| Doxycycline (DO,30) | S | 24 | S | 26 | S | 33 | S | 31 |
| Nalidixic acid (NA,30) | S | 21 | S | 15 | S | 29 | S | 25 |

S = Sensitive, R = Resistant, ZDI = Zone Diameter Inhibition value.

kanamycin (30µg/disc), neomycin (30 µg/disc), nalidixic acid (30 µg/disc), norfloxacin (10 µg/disc), ofloxacin (5 µg/disc), levofloxacin (5 µg/disc), rifampicin (5 µg/disc), tetracycline (30 µg/disc), streptomycin (10 µg/disc), vancomycin (30 µg/disc) (Table 10). Whereas all of the isolates except *Bacillus cereus* HABW1 (MN153450) was sensitive to amoxycillin (10 µg/disc). On the other hand, all the four bacterial isolates were found to be resistant towards standard dose of Penicillin (10 µg /disc) & ampicillin (10 µg/disc) (Table 10).

## Physiological tolerance of bacterial isolates

All four bacterial isolates could tolerate up to 4% NaCl concentration of the growth medium. All of the four isolates showed a wide range of pH tolerance (5–11), although they exhibited maximum amount of growth between pH 7.5–9.5 of the media (Fig 14). Growth of the bacterial isolates in the culture medium at different temperatures revealed that they could tolerate temperature range 15˚C-45˚C, although their growth became increased at 30˚C- 35˚C (Fig 15).

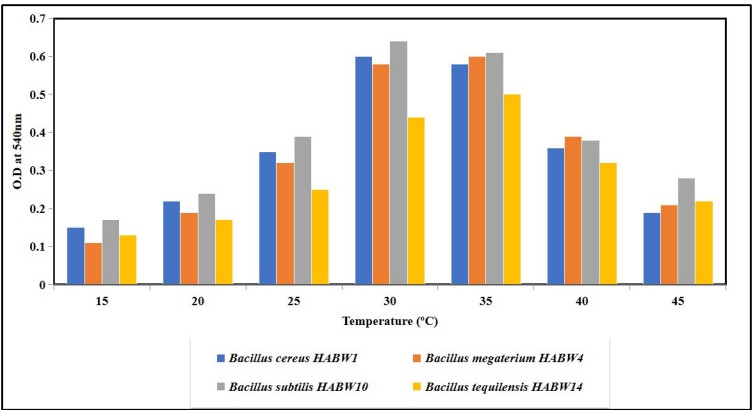

**Fig 14. Growth of bacterial isolates at different pH of the media.**

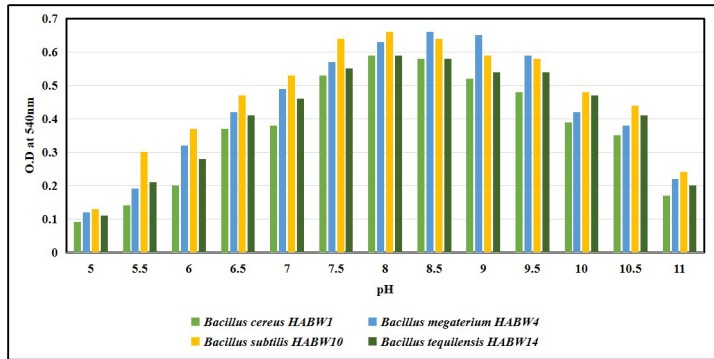

**Fig 15. Growth of bacterial isolates at different temperature of the media.**

## Discussion

Different species of mosquito prefer specific habitat water with diverse physicochemical characteristics for their egg laying and larval survival [11, 19, 47]. In natural environment some vector populations are found to be high in some aquatic habitats, while some others remain uncolonized, indicating that some places are more attractive for the gravid female mosquitoes than the others [48]. The selection of appropriate oviposition sites is very much crucial for the vector population dynamics and malarial epidemiology, as the immature vectors accomplish their life cycle and become adults within these preferred habitats [49, 50]. Physico-chemical properties, bacterial profile and organic matters of the breeding habitat water bodies are the key factors for the survivability of mosquito larvae [18, 51–53]. Selected environmental conditions provide specific harbourage sites of particular type of bacterial species to the inhabiting mosquitoes [22]. So, it is important to know about different factors which are influencing the daily oviposition pattern of gravid female vector mosquitoes for their successful control in the field. Several authors have studied the effect of physico-chemical and microbiological factors of breeding habitats, that either influence or deterrent different species of mosquito oviposition [16, 54].

The present study depicted that *An. subpictus* larval prevalence was comparatively higher in pond water than submerged rice-fields and drain water throughout all seasons of the year.

Rice-fields of study areas could not serve as potential breeding habitats for mosquito larvae because of the absence of water during winter season. Similar types of findings were reported by numerous researchers from different regions of the world [18, 55]. In some irrigated areas of Pakistan, prevalence of *An. subpictus* mosquitoes were reported to be higher than other some species of anopheline mosquitoes like *An. stephensi, An. culicifacies* & *An. pulcherrimus* in pond water rather than drain and irrigated fields [55]. Similarly, prevalence of *An. subpictus* mosquitoes were reported to be higher in pond water followed by rice-field water and the drain water in some malaria endemic areas of Bangladesh [18]. During the present study, although the higher larval density has been portrayed from the water logged ponds than submerged rice-fields, concordant similar physico-chemical parameters of the water bodies of ponds and rice-fields having some common oviposition attractant bacterial species indicated that there may be a possibility of the submerged rice-fields to be considered as major sources of *An. subpictus* mosquitoes in the absence of or much less number of ponds in an area. As the mosquitoes are poikilothermic animals, so their internal temperature fluctuates with the temperature of surrounding environment [56] and that is why environmental temperature have great impact in different life history stages of mosquitoes including egg hatching, larval development, emergence of adult mosquitoes and their subsequent vectorial capacity [57]. In the present study, prevalence of *An. subpictus* larvae were found to be higher in monsoon & post-monsoon season when the water temperature ranged between 27˚C -29˚C, whereas their occurrence became greatly reduced during winter season when the water temperature became reduced below 21˚C. This observation indicated that lower temperature of the surrounding environment might reduce the survival and development of mosquito larval population and thereby decreasing their prevalence. Similar type of observation was reported from some malaria endemic areas of Sri Lanka, where different anopheline mosquitoes including *An. subpictus* were recorded to be highly prevalent during monsoon and post-monsoon season [58]. Our observation showed analogy with the reports that in some regions of Himachal Pradesh, where *An. subpictus* larval abundance was found to be very low during the winter season [59].

Majority of anopheline species were recorded to prefer almost neutral pH of habitat water [47, 60], with some exceptions also [18, 61, 62]. According to the findings of the present study larval prevalence of *An. subpictus* was maximum in water bodies with pH range between 6.5–7.5, beyond which their prevalence gradually declined. Similar types of pH preference of *An. subpictus* mosquitoes were previously documented from some malaria endemic areas of Ratchaburi Province Thailand, where the pH of habitat water was found to range between 7.23–7.27 [60]. Present study delineated that larval density of *An. subpictus* had significant negative association with pH of habitat water when pH increased beyond 7.5. Several workers informed the negative correlation between anopheline larval abundance with pH of breeding habitat water [18, 60, 61], although some contrasting results were also reported [63]. Alike pH, in the study areas larval density of *An. subpictus* also showed negative correlation with the alkalinity of habitat water. Similar type of association was reported in *An. stephensi* mosquitoes from some rural and urban areas of West Bengal [64] although positive correlation was also documented in some areas of south-eastern Chennai [21].

*An. subpictus* mosquitoes in the study areas were found to prefer clear water with high dissolved oxygen (>5 mg/L) content for egg laying and showed significant positive correlation with amount of dissolved oxygen of habitat water. Several earlier workers have reported that most of the anopheline mosquito species favoured non-polluted water-bodies holding high dissolved oxygen content for egg laying and larval survival [12, 60, 64, 65], although contrasting results were documented in some regions of Pakistan as an uncommon event [13]. The preference for clear water is due to lack of siphon tube in anopheline larvae [66]. In the present study, *An. subpictus* larvae were found to be more prevalent in low turbid clear water than

highly turbid water bodies. Higher turbidity caused by the accumulation of both organic and inorganic compounds prevents the light penetration in the water bodies which might causes lesser number of photosynthetic organisms in the water bodies [63]. Lesser number of photosynthetic organisms lead to lower amount of dissolved oxygen content of water that might restrict the prevalence of *An. subpictus* larvae. Several workers reported that turbidity of water had a negative influence on the larval prevalence of different mosquito species like *Armigeres subalbatus, Culex quinquifasciatus, Aedes aegypti, Aedes albopictus, Toxorhynchites sp, Lutzia spp.* etc in some semi-urban and rural areas of Asam, India [67] though positive correlation was documented in case of *An. arabiensis* in some regions of Tubu village of Botswana [63].

The present study found that increase of water conductivity negatively affect the prevalence of *An. subpictus* larvae. Similar observation was reported in different mosquito species like *Armigeres subalbatus, Aedes albopictus, Aedes aegypti, Culex quinquifasciatus, Lutzia spp., Toxorhynchites sp.* etc from some rural and semi-urban areas of Asom, India [67]. *An. arabiensis* larvae in some regions of Tubu village, Botswana was also reported to have significant negative correlation with conductivity of habitat water [63]. In contrast, opposite result has also been reported by some workers such as, studies of Dida and coworkers reported positive association of both culicine and anopheline larvae with conductivity of breeding habitat water [12]. Conductivity of habitat water was also reported to have positive association with some other anopheline species like *An. stephensi* [21], *An. peryassui, An. albitarsis* and *An. nuneztovari* [68].

Present study observed that hardness of habitat water negatively affects the occurrence of *An. subpictus* larvae in the study areas. Similar findings were recorded in case of *Cx. quinquefasciatus* mosquitoes, where different degrees of water hardness found to have negative effect on the development of mosquitoes [69]. Hardness of water occurs due to the presence of dissolved minerals like magnesium, calcium etc. Sometimes high hardness values of water because of the presence of high amount of calcium ions may lead to the deposition of excess calcium within the cuticle of aquatic insects which may restrict expansion of cuticle, growth of insect and thus exert a toxic impact on them [69].

Amount of chloride ions present in water give indication of water salinity. In the present study chloride content of different water bodies were recorded to range between 34.38 mg/L-49.84 mg/L and no significant differences in the chloride content was observed between different habitats studied. Correlation study also indicated that prevalence of *Anopheles subpictus* larvae in different habitat water did not significantly influenced by the amount of chloride ion of water. Like the findings of the present study, amount of chloride ion of habitat water in some areas of south-east Iran was also reported to have no significant influence in the distribution of different species of anopheline larvae [51].

Present study found that concentration of nitrate ($NO_3^-$) and phosphate ($PO_4^-$) ions of habitat water had significant negative impact on the prevalence of *An. subpictus* larvae in the study areas. Nitrogen is one of the limiting factors for mosquito larval growth but excess concentration of nitrate in water might lead to eutrophication and exhaustion of dissolved oxygen content of water [70]. Similarly, high concentration of phosphate in water together with low dissolved oxygen content gives an indication of water pollution, that reported to have substantial negative effect on the longevity and body size of adult *Anopheles arabiensis* mosquitoes [71].

So, from the present study it is clear that, physico-chemical parameters of water are highly correlated with one another, such as increase in the dissolve ions and mineral contents of water increased the pH and hardness of water. Hard water with high pH might have negative impact on *An. subpictus* larvae. So, no single factor could play a major role in determining an ideal habitat, rather a group of factors together are responsible for generating a suitable habitat for the breeding and survival of *An. subpictus* mosquitoes. Through GLM analysis, it was

found that pH, alkalinity and dissolved oxygen content of water bodies are the major contributors for the variation of larval density of *An. subpictus* in different habitat types.

In addition to physico-chemical parameters of water bodies, several previous studies have indicated that microbial features of habitat water had also a great impact in determining larval density of different species of mosquitoes [28, 72]. During the present study, microbial analysis of different bacterial groups of breeding habitat water of *An. subpictus* mosquitoes indicated higher population of *Bacillus* group of bacteria with starch hydrolyzing & nitrate reducing capacity in the pond water, where the prevalence of *An. subpictus* larvae were recorded to be higher than rice-fields and drain water throughout the study period. *Bacillus* spp. could also enumerate in comparatively more clean, soft and stagnant breeding habitat waterbodies with high dissolve oxygen content, where *An. subpictus* mosquitoes also showed higher prevalence exhibiting the co-existence of anopheline larvae and spore forming *Bacillus* bacterial strains. Some earlier studies had reported *Bacillus* group of bacteria have the ability to modulate different physico-chemical parameters of water and thus making it favourable for many aquatic organisms. For instances, studies by some workers showed that *Bacillus* group of bacteria including *Bacillus megaterium*, *Bacillus subtilis* and *Bacillus licheniformes* together played some important role which improves the dissolved oxygen content of water [73, 74]. In addition to that *Bacillus* group of bacteria were also found to maintain the alkalinity and pH of water bodies and avert it from becoming too low or too high [75]. Higher value of total dissolved solids (TDS) of water owing to pollution might have adverse effect on aquatic organisms. Several *Bacillus* groups of bacteria including *Bacillus subtilis*, *Bacillus pumilus*, *Bacillus licheniformis*, *Bacillus cereus* had been reported to maintain TDS values of water within tolerable range that improves quality of water [76, 77]. Higher amount of nitrate and phosphate content of water are the reasons for algal bloom formation which ultimately reduced the quality of water. *Bacillus* group of bacteria including *Bacillus cereus*, *Bacillus mojavensis*, *Bacillus subtilis* had been reported to reduce nitrate [24, 78] and phosphate [79] content of water and thus helped in improving the water quality. Several *Bacillus* spp. like *B. subtilis*, *B. cereus*, *B. mojavensis* were reported to have the ability to reduce total hardness values of water [79, 80]. *Bacillus* group of bacteria also have the capacity to decompose organic materials present in pond water to smaller units and thus serve to improve the water quality [81–83]. Higher starch hydrolyzing and nitrate reducing bacterial populations in habitat water help to degrade starch content of water bodies and reduce nitrate level of water. Earlier studies reported *Bacillus cereus* XHJ-2-6 present in shrimp pond water had the ability to reduce total suspended solids of water by its proteolytic and amylolytic activity which improves the quality of water [84]. All these reports corroborated the outcomes of the present study in a way that, higher population of *Bacillus*, starch hydrolyzing & nitrate reducing bacteria in water bodies might have a role to improve the water quality parameters. The microbial metabolic and physiological activities in larval habitat water modified several water physico-chemical parameters and created a suitable habitat condition for *An. subpictus* larvae. Thereby both the *Bacillus* group of bacteria and *An. subpictus* showed a co-existence in these water bodies.

The present microbiological study of anopheline breeding habitats recorded eight bacterial strains were common in all habitat types of *An. subpictus* mosquito throughout the year. Oviposition study in laboratory condition revealed that, among these eight isolates only four acted as potent attractant of mosquito oviposition, whereas other four isolates did not have any significant influence on mosquito oviposition. Morphological, bio-chemical and molecular analyses confirmed that these four bacterial strains were different species of *Bacillus* viz., *B. cereus* HABW1, *B. megaterium* HABW4, *B. subtilis* HABW10 & *B. tequilensis* HABW14. Like the findings of the present study, Mondal and his co-workers also reported four common bacterial isolates identified as *Bacillus* sp. from all types of mosquito larval habitats in Dehradun City of

Uttarakhand [53]. Oviposition study in laboratory condition revealed that although these four bacterial species significantly influenced the mosquito oviposition but the attractancy rate varied. Oviposition activity index (OAI) was recorded as 0.79, 0.62, 0.80 & 0.62 towards *B. cereus* HABW1, *B. megaterium* HABW4, *B. subtilis* HABW10 & *B. tequilensis* HABW14 respectively, which indicated that *B. subtilis* HABW10 and *B. cereus* HABW1 were more potent attractant of gravid *An. subpictus* mosquitoes than *B. megaterium* HABW4 and *B. tequilensis* HABW14. Earlier studies by several workers also indicated that not all bacteria inhabiting in the breeding habitat of mosquito have positive influence on mosquito oviposition, further some of them also repel the gravid mosquito oviposition, such as *Anopheles gambiae* mosquitoes in Kenya were reported to lay lower amount of eggs in water containing a mixture of different bacterial isolates of natural habitat including *Bacillus*, *Enterobacter*, *Aeromonas*, *Stenotrophomonas*, *Acinetobacter*, *Klebsiella* and *Pseudomonas* than bacteria free control water. In addition they also recorded that bacterial isolate *Stenotrophomonas maltophilia* repel oviposition of gravid *Anopheles gambiae* mosquitoes [85]. Similarly, Lindh and coworkers reported among seventeen bacterial isolates (eight from the mid-gut of *Anopheles gambiae* and nine from the breeding habitat water), gravid *Anopheles gambiae* mosquito had positive ovipositional response to six bacterial isolates. Among these six isolates, five were from the breeding habitat (*Bacillus* sp., *Comamonas* sp., *Proteus* sp., *Exiguobacterium* sp. and *Micrococcus* sp.) and one (*Vibrio metschnikovii*) from the mid-gut of *Anopheles arabiensis* mosquito [86].

Earlier workers elicited the variations in microbial composition among mosquito larvae prevailing in different habitats, but harmony and propinquity of the same sharing the same habitat [87]. During the present study, analyses of different bacterial groups indicated the abundance of spore forming *Bacillus* population in the pond water with the higher prevalence of *An. subpictus* larvae than those occurring in rice-fields and drain water throughout the entire study period. Present study identified four bacterial strains, namely *Bacillus cereus* HABW1 (MN153450), *B. megaterium* HABW4 (MN173350), *B. subtilis* HABW10 (MN166905) and *B. tequilensis* HABW14 (MZ363639) and all of them except *B. megaterium* HABW4 (MN173350) were positive for starch hydrolysis and nitrate reduction test. So, it may be inferred that spore forming *Bacillus* group having starch hydrolyzing, nitrate reducing properties, were present in higher frequency in pond water than rice-field and drain water, and acted as potent oviposition attractant strains for gravid female *An. subpictus*.

Oviposition attractancy of gravid female mosquitoes is due to some volatile chemicals that are released through bacterial metabolic activities [86, 88]. A species specific variation in respect to these volatile chemicals was also noticed [89]. Earlier observation documented the production of volatile chemicals due to bacterial fermentation of different organic matters of habitat water. Studies by Santana and associated workers reported that, *Aedes aegypti* mosquitoes were attracted to microbial volatile released due to fermentation of *Panicum maximum* grass by the microbial activities [90]. Present study recorded that the identified oviposition attractant bacterial isolates could ferment a good number of carbohydrate sources, which indicate their high fermentation capabilities. In addition to that, all of them except *Bacillus megaterium* HABW4 was positive for protein hydrolysis & starch hydrolysis test which indicated the ability of these bacterial species to degrade protein & starch content of animal or plant origin, and thus could contribute in the decomposition of organic materials present in the water bodies. Physiological tolerance test revealed that all of these four-oviposition attractant bacterial strains could tolerate a wide range of pH and temperature of the environment, which helped them to survive in adverse environmental condition and all of them could have a tolerance up to 4% NaCl concentration of the growth media, which indicated that they could survive in slightly saline environment also. Present observation depicted that higher larval prevalence of *An. subpictus* at a water temperature ranging between 27˚C-29˚C and pH of the

habitat water ranging between 6.5–7.5. In these temperature and pH range, breeding habitat bacteria also showed higher amount of growth. Although antibiotic sensitivity tests revealed the sensitivity of all the isolates to most of the standard antibiotics, still application of antibiotics to natural breeding habitats might have several harmful effects on environment [91, 92]. Therefore more eco-friendly approach needs to be explored for effective control of these oviposition attractant bacterial strains from mosquito breeding habitats as a great option of malaria management programme.

## Conclusion

Present investigation potrays that *An. subpictus* mosquitoes prefer to breed in non-polluted clear water bodies having higher amount of dissolved oxygen content and their prevalence becomes greatly augmented during monsoon and post-monsoon season than summer and winter. So, besides polluted water we should turn from our contemplation of the non-polluted water bodies. Populations of different microbial groups might help to modulate the physico-chemical parameters of water and thus making it more suitable for *An. subpictus* mosquitoes. During the present study four bacterial strains *Bacillus cereus* HABW1, *Bacillus megaterium* HABW4, *Bacillus subtilis* HABW10 and *Bacillus tequilensis* HABW14 were identified as potent ovipositional attractants of the gravid *An. subpictus* mosquitoes prevalent in rural areas of Hooghly District, West Bengal, India. Further elucidation about the microbial activities contributing to the favourable environment for oviposition and vector survival might improve the current strategies of vector management programme. If these oviposition attractant bacterial isolates could be ruined from the mosquito breeding sites through ecofriendly bactericidal or bacteriostatic plant extracts, the rate of egg laying by mosquito vectors will be minimized, in such a way, which would obviously contribute an alternative strategy of vector management in malaria prone areas.

## Supporting information

**S1 File. Data of larval density and physico-chemical parameters of different types of aquatic bodies in four different seasons.**
(XLSX)

**S2 File. Data of number of colonies counted over respective agar media and cfu of different bacterial populations in different types of aquatic bodies in four different seasons.**
(XLSX)

**S3 File. Data related to mosquito oviposition.**
(XLSX)

**S1 Fig. Growth of bacterial colonies of mosquito breeding habitats on different agar media.**
(TIF)

**S2 Fig. Gram staining of common bacterial isolates from breeding habitats of *An. subpictus* mosquito.**
(TIF)

**S1 Table. A & B. Friedman test for significant effect of season and habitat types on larval density of *Anopheles subpictus*.**
(DOCX)

**S2 Table. One-Way ANOVA for physico-chemical parameters of different habitat types (ponds, drains & rice-fields) during summer season.**
(DOCX)

**S3 Table. One-Way ANOVA for physico-chemical parameters of different habitat types (ponds, drains & rice-fields) during monsoon season.**
(DOCX)

**S4 Table. One-Way ANOVA for physico-chemical parameters of different habitat types (ponds, drains & rice-fields) during post-monsoon season.**
(DOCX)

**S5 Table. Mann-Whitney test for physico-chemical parameters between ponds and drains during winter season.**
(DOCX)

**S6 Table. A, B & C. Principal Component Analysis (PCA) for larval density and physico-chemical parameters of habitat water during summer season.**
(DOCX)

**S7 Table. A, B & C. Principal Component Analysis (PCA) for larval density and physico-chemical parameters of habitat water during monsoon season.**
(DOCX)

**S8 Table. A, B & C. Principal Component Analysis (PCA) for larval density and physico-chemical parameters of habitat water during post-monsoon season.**
(DOCX)

**S9 Table. A, B & C. Principal Component Analysis (PCA) for larval density and physico-chemical parameters of habitat water during winter season.**
(DOCX)

**S10 Table. A, B & C. Univariate Tests of Significance for larval density (L.D).**
(DOCX)

## Acknowledgments

The authors are thankful to the Burdwan University authority for providing proper laboratory facilities to carry out the work. The authors are thankful to Dr. Ayan Mondal, Assistant professor, Government General Degree College, Mohanpur for his constructive suggestions in statistical analyses. Authors are also very much thankful to DST PURSE, DST FIST for providing instrumental facilities.

## Author Contributions

**Conceptualization:** Soumendranath Chatterjee.

**Data curation:** Madhurima Seal.

**Formal analysis:** Madhurima Seal.

**Investigation:** Madhurima Seal.

**Supervision:** Soumendranath Chatterjee.

**Visualization:** Soumendranath Chatterjee.

**Writing – original draft:** Madhurima Seal.

**Writing – review & editing:** Soumendranath Chatterjee.

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
