## [Decision Letter · Decision Letter 0]

27 Sep 2022

PONE-D-22-19958Combined effect of physico-chemical and microbial quality of breeding habitat water on oviposition of Anopheles subpictus  in Hooghly, West Bengal, India.PLOS ONE

Dear Dr. Chatterjee,

Thank you for submitting your manuscript to PLOS ONE. After careful consideration, we feel that it has merit but does not fully meet PLOS ONE’s publication criteria as it currently stands. Therefore, we invite you to submit a revised version of the manuscript that addresses the points raised during the review process.

Both reviewers highlighted common faults, such as the need to improve statistical analyses and to perform a general reorganization of the data presented.

Additional comments from each of them should also be carefully considered when submitting the revised version, in order to improve the readability of the manuscript.

We look forward to receiving your revised manuscript.

Kind regards,

Cinzia Calvio, PhD

Academic Editor

PLOS ONE

Journal Requirements:

“The authors are thankful to University Grant Commission 783 (UGC, India) and WBDSTBT, Govt. of West Bengal for providing financial support.”

6. We note that Figure 1 in your submission contain satellite images which may be copyrighted. All PLOS content is published under the Creative Commons Attribution License (CC BY 4.0), which means that the manuscript, images, and Supporting Information files will be freely available online, and any third party is permitted to access, download, copy, distribute, and use these materials in any way, even commercially, with proper attribution. For these reasons, we cannot publish previously copyrighted maps or satellite images created using proprietary data, such as Google software (Google Maps, Street View, and Earth). For more information, see our copyright guidelines: http://journals.plos.org/plosone/s/licenses-and-copyright.

Reviewers' comments:

Reviewer's Responses to Questions

**Comments to the Author**

1. Is the manuscript technically sound, and do the data support the conclusions?

Reviewer #1: Yes

Reviewer #2: Yes

2. Has the statistical analysis been performed appropriately and rigorously? 

Reviewer #1: No

Reviewer #2: No

3. Have the authors made all data underlying the findings in their manuscript fully available?

Reviewer #1: Yes

Reviewer #2: Yes

4. Is the manuscript presented in an intelligible fashion and written in standard English?

Reviewer #1: Yes

Reviewer #2: No

5. Review Comments to the Author

Reviewer #1: I have read with interest this manuscript. The study is well designed, and the conclusions are supported (albeit not fully, see comments below), but I have some issues that I think the authors should address when preparing a new version of the manuscript. The main scope, at least if I understood correctly, was to identify the biotic (bacteria) and abiotic factors that are associated to oviposition choice (and larval presence/abundance). However, the study, contains many analyses that do not seem to be fully relevant to the aim of the study and sometimes it feels like the authors packed many experiments without a clear idea on their utility. E.g.: why performing electron microscopy analyses? Why antibiotic sensitivity? Why metabolic and physiological experiments? The authors should make clearer on what the entire work is based on how all such analyses are related.

I also provide a list of comments that I hope the authors would find helpful in preparing the revised version of their work.

Title: I think that it is not necessary to include the experimental locality in the title

Abstract: the authors should remove the detailed results from the abstract and instead report a concise overview on what are the main findings.

Introduction: in general, authors should 1) provide more information on the biology of Anopheles subpictus and its role in malaria transmission; 2) explain hypotheses and aims.

Line 50: I would use “species” instead of “genus and species”

Line 62: the authors say that “some had no correlation with the larval abundance”, however all previous lines refer to parameters that correlate to other factors, so it is not clear why mentioning non correlation to just larval abundance.

Lines 74-78: I think that the manuscript would benefit from referring to the experimental locality as a case study whose results can have a broader impact. Here it reads as if the experiments and their results are restricted and relevant to Hooghly mosquitoes only.

Lines 74-78: there is a discrepancy between the “aim” of knowing more the breeding habitat of the anopheline fauna and the fact that you then focus on An. subpictus.

Lines 88-92: how many replicas per aquatic body type?

Line 108: define S.E. (not at line 128)

Line 115: define BOD

Line 132: what does 10e-3 refers to? Dilution factor?

Line 154: are the breeding habitats like those used for bacterial sampling? Fort instance, can you exclude the possibility that bacterial composition of the sampling site did not interfere with oviposition site choice during the experiments? (bacteria experienced during the larval stage may for example influence the preference /avoidance for certain bacterial type present in the oviposition water)

Line 161: was the water from the natural breeding habitat from which that particular strain was collected or from a “general” habitat?

Lines 173-…: please clarify which bacterial isolates were used for the inoculations and where they come from.

Line 214: I do not understand what measure was done and what “nearly similar bacteria” refers to.

Line 215: NJ phylogenies are not reliable, if possible I highly recommend performing a ML (e.g using RAxML) analysis instead.

Lines 228-244: not clear why such experiments and analyses were conducted, please add one line to briefly explain the reason why antibiotic sensitivity and carbohydrate fermentation were necessary for the overall study.

Lines 247-249: please add some details about the experimental procedure.

Line 250 (Statistical analysis section): why not using a Tukey also for the differences of physico-chemical parameters among habitat types? Also, I recommend a non-parametric test (e.g. Wilcoxon) instead of the t-test

Table 1: I think that for the kind of data you are reporting, the use of Standard Deviation instead of S.E. is more suited.

Lines 279-…: I think that the series of numbers reported in this section are very difficult to interpret and that what reported in table 2 suffice for the reader, unless the authors prefer to point out some significant differences between habitats/season.

Lines 308-311 (and afterwards): please specify the test used to obtain the P values

Lines 338-…: please report P values associate to the Person correlation coefficients: negative or positive correlations, unless supported by statistically significant P values have no meaning; since all such r values are reported in Table 3 it is not necessary to repeat them in the main text (and I think a multiple testing correction would be desirable in your case, since many tests were performed; e.g. Holms-Bonferroni correction)

Line 360: what does it mean “significant correlation at alpha=0.05, p=<0.0001”?

Lines 363-..: as before, leave all numerical results in the Table and rather report only significant results.

Lines 391-…: why these bacteria isolates were not characterized using molecular approaches (see lines 194-215)? How much the use of culture-dependent identification methods may have affected the results?

Lines 400-…: again, you have already reported all results in the table, no need to repeat them here

Lines 427-…: again, you have already reported all results in the table, no need to repeat them here

Lines 488-490: these results are not relevant to the scope of the study; I think it is not necessary to report them

Lines 491-496: bootstraps values <80 or <90, depending on how much you wish to be conservative, are not reliable and therefore should not interpreted as a robust result (please write without %). I think it also more correct to write “X is closely related to Y” rather than “X branched with Y”. The phylogenetic analyses in this case are informative only regarding the species identification.

Lines 507-…: not sure I understand the usefulness of these analyses and results.

Lines 578-579: can the authors say something about the bacterial diversity present in the different habitats, so to make a connection to their analyses on oviposition choice driven by bacterial strain prevalence?

Lines 578-587: larval density is higher in ponds than rice fields, however can the authors say something about the extension of the two habitats and therefore on their role as breeding sites for Anopheles? For instance, if ponds are much less in umber and rice fields have a much larger extension, the latter may however represent the major source of mosquitoes.

Lines 588-599: not clear why temperature is considered as a major determinant of breeding success, since, as the authors have shown, other factors differ from monsoon and non-monsoon seasons.

Line 603: how much pH, and other abiotic factors, can affect bacterial composition and vice versa? In other words, what is the driving factor affecting other variables considered in this (and other) studies on habitat suitability?

Lines 613-616: is there the possibility that the contrasting preference between pH in the two species is the result of niche partitioning?

Line 625: without time-series data it is very speculative to claim possible adaptation to polluted water bodies, as they may just represent cases of oviposition site choice at the boundaries of the species’ preference.

Line 640: in general, I think that many factors that contribute to larvae abundance and oviposition site choice may be highly correlated one another. You should for example refer to the PCA plots, where you indicated (arrows) which factors co-correlate with the PCA components.

Lines 707-709: please discuss the possibility that both Bacillus and Anopheles prefer a certain type of water properties and thus the presence of the former is not a condition but a coincidence with the presence of the latter. (see also next section; and, in general, check this reference: https://www.frontiersin.org/articles/10.3389/fmicb.2019.02036/full - it refers to Aedes species but may contain some useful information for your study)

Lines 752-761: the tolerance displayed by bacteria may not be relevant for mosquito control if, as the authors showed in this MS, mosquitos have a narrower tolerance towards such abiotic factors.

Lines 776-777: can the authors elaborate a little on which kind of bacterial manipulation can be performed to control mosquitoes?

Reviewer #2: The authors on this paper describe the physio-chemical features of the oviposition sites and the microbial community that might be linked to the attraction of female mosquitoes of Anopheles sub-ictus to the sites.

The paper is scientifically sound and the findings are interesting, but I think some points need clarification/revision.

STATISTICAL ANALYSIS

The analysis of number of larvae per season in the different ponds have been done using two-way ANOVA. However, giving that the Rice-field are empty during winter, the data are missing (resulting in a non-normal data). This is a strong violation to the ANOVA requirements (there is no standard deviation, and the ANOVA evaluates the variation of the variance). I think a non-parametric test is more appropriate or a different statistical approach.

Analysis of the physio-chemical features on the oviposition sites. The authors used PCA followed by a regression analysis. I do not agree for two reasons: first, apart reporting the r value they should report the associated P. Second: doing that, they are testing multiple variables on the same outcome multiple times and that increase the Type I error in the analysis. I think a more feasible approach is the construction of a model (general linearized or mixed model?) in which all the variables are modelled for their influence on the outcome (the number of larvae).

Student t test of the number of eggs laid using isolates. I do not understand why the authors used an unpaired student t test instead of a paired t. The pairing is the two oviposition cups from the same experiment (those that contained water and isolates).

REPORT OF THE RESULTS.

I suggest the authors to remove from the text (and also from the abstract) the description of the results that are already reported in the tables. It does not add anything to the paper, is not easy to read and to interpret and the data are available in the tables.

MICROBIOLOGY ANALYSIS.

At line 363, in the results. I would avoid calling it "bacteria diversity". There are more sophisticated analysis of diversity and reducing bacteria diversity to the cultivable portion is strongly reductive. Furthermore, I think it is not the focus of the authors. The point here is to find isolates common to all the riding sites that can function ass attractants. I would re-write the paper in this terms, as otherwise is not clear.

METHODS:

Line 87: the authors state that they counted the larvae per dip. How were they sure they are An. subpictus? They mentioned later (in the oviposition assay) that they collected larvae, brought them in the lab and selected An. subpictus for oviposition assays, which means multiple species are present. Did they check all the larvae they've collected?

English needs revision. For example, at line 573, "It is very much important"...I would avoid such an hyperbolic phrasing.

6. PLOS authors have the option to publish the peer review history of their article (what does this mean?). If published, this will include your full peer review and any attached files.

Reviewer #1: No

Reviewer #2: No

---

## [Author Response · Author response to Decision Letter 0]

12 Nov 2022

To

Cinzia Calvio

Academic Editor

PLoS ONE

Sub: Submission of revised manuscript (PONE-D-22-19958)

Dear Sir,

Thank you for your email dated 28/09/2022 enclosing the reviewers’ comments. We are very much thankful to you and to the hon’ble reviewers for their valuable comments and constructive suggestions. We have carefully reviewd all the comments and have revised the manuscript accordingly. Our responses are given point-by-point manner below and the modified portions have been highlighted in the revised manuscript. 

We look forward to hearing from you in due course.

With regards,

Dr. Soumendranath Chatterjee

Professor

Department of Zoology

The University of Burdwan

Golapbag, Burdwan, West Bengal

Our manuscript meets PLOS ONE's style requirements, including those for file naming. If there are any mistake, please let us know, so that we can modify that portion.

According to your suggestion we have removed figure 1 from the revised manuscript as we were not able to take permission from the appropriate authority. 

We have provided the minimal data set underlying the results described in our manuscript and we have included the captions in Supporting Information files at the end of our manuscript, and also referred in-text to match accordingly.

Reviewer#1

Comment 1: Title: I think that it is not necessary to include the experimental locality in the title.

Author’s response: According to the suggestion of hon’ble reviewer the name of experimental regional locality is omitted from the title.

Comment 2: Abstract: the authors should remove the detailed results from the abstract and instead report a concise overview on what are the main findings.

Author’s response: The abstract portion has been modified as per the suggestion of hon’ble reviewer and concise overview of the results has been incorporated in the abstract.

Comment 3: Introduction: in general, authors should 1) provide more information on the biology of Anopheles subpictus and its role in malaria transmission; 2) explain hypotheses and aims.

Author’s response: 1. According to the suggestion of hon’ble reviewer, information on the biology of Anopheles subpictus and its role in malaria transmission have been provided in the revised manuscript. (Line no. 49-55 in revised manuscript).

2.The hypotheses and aims of the study have been included as per the suggestion of the hon’ble reviewer: “Our study focused whether both biological and chemical breeding habitat parameters have influenced the propagation and multiplication of this particular rural malarial vector species. So, the present study has been aimed to determine the significant physico-chemical characteristics as well as the microbial markers with a special reference to oviposition attractant bacterial strains having a positive influence towards the oviposition behaviour of gravid female An. subpictus mosquitoes”. (Line no. 80-86 in revised manuscript).

Comment 4: Line 50: I would use “species” instead of “genus and species”

Author’s response: According to the suggestion of hon’ble reviewer the word has been modified (Line no. 56 in revised manuscript).

Comment 5: Line 62: the authors say that “some had no correlation with the larval abundance”, however all previous lines refer to parameters that correlate to other factors, so it is not clear why mentioning non correlation to just larval abundance.

Author’s response: Reports of several previous research studies have indicated that among different physico-chemical parameters of mosquito breeding habitats, some parameters showed strong correlation (positive or negative) with larval pevalence whereas some had no correlation at all. That’s why this sentence was written so. This sentence has been modified: “Some of these parameters showed strong positive correlation, with the larval abundance” (Line no. 67-68 in revised manuscript).

Comment 6: Lines 74-78: I think that the manuscript would benefit from referring to the experimental locality as a case study whose results can have a broader impact. Here it reads as if the experiments and their results are restricted and relevant to Hooghly mosquitoes only.

Author’s response: From the title as well as from the manuscript the name of the regional experimental locality has already been omitted. An. subpictus mosquitoes is a house frequenting mosquito species and its distribution has been recorded from maximum countries of the world. It’s co-existance with other mosquito species in different breeding habitats has also been recorded by various researchers all over the world. So, although the study has been done in some rural areas of India, but the results of this study have really a broader impact which can elucidate the control strategies to be taken in other areas also. 

Comment 7: Lines 74-78: there is a discrepancy between the “aim” of knowing more the breeding habitat of the anopheline fauna and the fact that you then focus on An. subpictus.

Author’s response: This portion have been corrected in the revised manuscript (Line no. 80 in revised manuscript).

Comment 8: Lines 88-92: how many replicas per aquatic body type?

Author’s response: Twenty replicas were taken per aquatic body type (Line no. 100 in revised manuscript). 

Comment 9: Line 108: define S.E. (not at line 128)

Author’s response: S.E. is defined in revised manuscript (Line no. 122 in revised manuscript).

Comment 10: Line 115: define BOD

Author’s response: BOD is defined in revised manuscript (Line no. 129 in revised manuscript).

Comment 11: Line 132: what does 10e-3 refers to? Dilution factor?

Author’s response: 10-3 refers dilution (Line no. 147 in revised manuscript).

Comment 12: Line 154: are the breeding habitats like those used for bacterial sampling? For instance, can you exclude the possibility that bacterial composition of the sampling site did not interfere with oviposition site choice during the experiments? (bacteria experienced during the larval stage may for example influence the preference /avoidance for certain bacterial type present in the oviposition water)

Author’s response: Yes, same mosquito breeding habitats were used for physico-chemical and bacterial sampling from where the larval collection were done, as bacterial composition definitely interferes with the oviposition of gravid female mosquitoes (Line no. 103-106 in revised manuscript).

Comment 13: Line 161: was the water from the natural breeding habitat from which that particular strain was collected or from a “general” habitat?

Author’s response: The oviposition attractant bacterial strains were isolated from the natural breeding habitat water bodies of An. subpictus mosquitoes. 

Comment 14: Lines 173-…: please clarify which bacterial isolates were used for the inoculations and where they come from.

Author’s response: Resident bacterial isolates (HABW1, HABW3, HABW4, HABW6, HABW10, HABW12, HABW14 and HABW15), prevalent all through the year in the natural breeding habitat water bodies of An. subpictus mosquitoes were exploited for the oviposition studies. (Line no. 186-188 in revised manuscript).

Comment 15: Line 214: I do not understand what measure was done and what “nearly similar bacteria” refers to.

Author’s response: Phylogenetic affiliation of the bacterial strains have been written as per the modified phylogenetic trees. ML phylogenetic analysis have also performed.

Comment 16: Line 215: NJ phylogenies are not reliable, if possible I highly recommend performing a ML (e.g using RAxML) analysis instead.

Author’s response: Phylogenetic affiliation of the bacterial strains have been written as per the modified phylogenetic trees. Phylogenetic trees performed through ML analysis method have also included in the revised manuscript (Line no. 227- 228, 502-524, Figs. 8-11).

Comment 17: Lines 228-244: not clear why such experiments and analyses were conducted, please add one line to briefly explain the reason why antibiotic sensitivity and carbohydrate fermentation were necessary for the overall study.

Author’s response: Release of oviposition attractant volatiles is associated with bacterial fermentation of different carbohydrate sources. So, in the laboratory we performed fermentation tests of bacterial isolates to different carbohydrate sources in addition to characterization purpose of the isolates. Besides characterization purpose, sensitivity of oviposition attractant bacterial strains to recommended doses of commercially available antibiotics was checked to determine the potent antibiotics which could have a significant antibacterial sensitive zone as recommended by the disc diffusion method and which could have an ability to kill or eliminate those particular bacterial strains performing as microbial markers for the oviposition of gravid female An. subpictus mosquitoes in their natural breeding habitat water bodies. Explanations for these tests have been added in the revised manuscript (Line no. 243-244, 259-263).

Comment 18: Lines 247-249: please add some details about the experimental procedure.

Author’s response: According to the suggestion of hon’ble reviewer, details about the experimental procedure of physiological tolerance test have been added in the revised manuscript. (Line no. 268-278 in revised manuscript).

Comment 19: Line 250 (Statistical analysis section): why not using a Tukey also for the differences of physico-chemical parameters among habitat types? Also, I recommend a non-parametric test (e.g. Wilcoxon) instead of the t-test. 

Author’s response: According to the suggestion of hon’ble reviewer, Tukey test was performed to evaluate the differences of physico-chemical parameters among habitat types in different seasons and the results have been added in the revised manuscript (Line no. 281-282 and Figs 2-5 in revised manuscript). According to recommendation of hon’ble reviewer a non-parametric test (Mann-Whitney test) was performed instead of t-test for winter season (Line no. 284, 337-339, 341-342 and Table S5 ).

Comment 20: Table 1: I think that for the kind of data you are reporting, the use of Standard Deviation instead of S.E. is more suited.

Author’s response: According to the suggestion of hon’ble reviewer, in table 1, while reporting the result of per dip larval density, Standard Deviation (S.D) was used instead of standard error (S.E). (Line no. 310-312, Table 1)

Comment 21: Lines 279-…: I think that the series of numbers reported in this section are very difficult to interpret and that what reported in table 2 suffice for the reader, unless the authors prefer to point out some significant differences between habitats/season.

Author’s response: According to the suggestion of hon’ble reviewer, data series from the text have been removed and presented only in the table.

Comment 22: Lines 308-311 (and afterwards): please specify the test used to obtain the P values

Author’s response: According to the suggestion of hon’ble reviewer, the test used to obtain the p values have been specified in the revised manuscript. (Line no. 333-345 in revised manuscript).

Comment 23: Lines 338-…: please report P values associate to the Person correlation coefficients: negative or positive correlations, unless supported by statistically significant P values have no meaning; since all such r values are reported in Table 3 it is not necessary to repeat them in the main text (and I think a multiple testing correction would be desirable in your case, since many tests were performed; e.g. Holms-Bonferroni correction).

Author’s response: As per the suggestion of hon’ble reviewer, multiple testing correction (Holms-Bonferroni correction) have been performed and the p values associate to the Person correlation coefficients have been given in the revised manuscript (Table 3).

Comment 24: Line 360: what does it mean “significant correlation at alpha=0.05, p=<0.0001”?

Author’s response: This portion have been corrected in the revised manuscript.

Comment 25: Lines 363-..: as before, leave all numerical results in the Table and rather report only significant results.

Author’s response: According to the suggestion of hon’ble reviewer, this portion have been modified in revised manuscript.

Comment 26: Lines 391-…: why these bacteria isolates were not characterized using molecular approaches (see lines 194-215)? How much the use of culture-dependent identification methods may have affected the results?

Author’s response: All resident bacterial isolates were characterized through phenotypic and bio-chemical methods for primary screening. Molecular characterization with a special reference to phylogenetic affiliation based on 16S rRNA gene sequencing of oviposition attractant bacteria were performed besides their phenotypic and bio-chemical characterizations for the identification through polyphasic taxonomic methods of the particular strains responsible for the oviposition of An. subpictus mosquito.

Comment 27: Lines 400-…: again, you have already reported all results in the table, no need to repeat them here

Author’s response: According to the suggestion of hon’ble reviewer, data (provided in table 5) have been removed from the text. (Line no. 454-455 in revised manuscript).

Comment 28: Lines 427-…: again, you have already reported all results in the table, no need to repeat them here

Author’s response: According to the suggestion of hon’ble reviewer, data (provided in table 7) have been removed from the text.

Comment 29: Lines 488-490: these results are not relevant to the scope of the study; I think it is not necessary to report them. 

Author’s response: According to the suggestion of hon’ble reviewer, results of this section have been omitted from the revised manuscript.

Comment 30: Lines 491-496: bootstraps values <80 or <90, depending on how much you wish to be conservative, are not reliable and therefore should not interpreted as a robust result (please write without %). I think it also more correct to write “X is closely related to Y” rather than “X branched with Y”. The phylogenetic analyses in this case are informative only regarding the species identification.

Author’s response: According to the suggestion of hon’ble reviewer, this portion have been modified in revised manuscript. (Line no. 227- 228, 502-524, Figs. 8-11).

“Through neighbour-joining method Bacillus cereus HABW1 (MN153450) was found closely related to B. cereus (MH210863), whereas, ML method indicated that the bacterial isolate B. cereus HABW1 (MN153450) was closely similar to B. cereus (HQ684015). Neighbour-joining tree of Bacillus megaterium HABW4 (MN173350) indicated that this bacteria is closely related with B.megaterium (KX495254) and according to ML tree this bacterial isolate is closely related with B. megaterium (KP017584) & B. megaterium (HQ634276).Both neighbour-joining and ML tree of Bacillus subtilis HABW10 (MN166905) indicated that this bacterial strain is closely related to B. subtilis (EF633176). Phylogenetic tree prepared by both neighbour- joining and ML method indicated that Bacillus tequilensis HABW14 (MZ363639) closely related with B. tequilensis (MK018119)”

Comment 31: Lines 507-…: not sure I understand the usefulness of these analyses and results.

Author’s response: Through scanning electron microscopy bacterial shape, surface morphology and endospore bearing properties were examined. 

Comment 32: Lines 578-579: can the authors say something about the bacterial diversity present in the different habitats, so to make a connection to their analyses on oviposition choice driven by bacterial strain prevalence?

Author’s response: Reports of earlier workers elicited the differences in microbial composition among mosquito larvae present in different habitats but similar bacterial communities among larvae sharing the same breeding habitat (Scolari et al, 2019). During the present study, analyses of different bacterial groups indicated the abundance of spore forming Bacillus population with starch hydrolyzing & nitrate reducing capacity in the pond water, with the higher prevalence of An. subpictus larvae than those occurring in rice-fields and drain water throughout the entire study period. Present study identified four bacterial strains, namely Bacillus cereus HABW1 (MN153450), B. megaterium HABW4 (MN173350), B. subtilis HABW10 (MN166905) and B. tequilensis HABW14 (MZ363639) and all of them except B. megaterium HABW4 (MN173350) were positive for starch hydrolysis and nitrate reduction test. So, it may be inferred that spore forming Bacillus group having starch hydrolyzing, nitrate reducing properties, were present in higher frequency in pond water than rice-field and drain water, and acted as potent oviposition attractant strains for gravid female An. subpictus. This portion is discussed in the revised manuscript (Line no. 756-767 in revised manuscript).

Comment 33: Lines 578-587: larval density is higher in ponds than rice fields, however can the authors say something about the extension of the two habitats and therefore on their role as breeding sites for Anopheles? For instance, if ponds are much less in number and rice fields have a much larger extension, the latter may however represent the major source of mosquitoes.

Author’s response: Although the higher larval density has been potrayed from the water logged ponds than submerged rice-fields, the similar physico-chemical parameters of the water bodies of ponds and rice-fields having some common oviposition attractant bacterial strains indicate that there may be a possibility of the submerged rice-fields to be considered as major sources of An. subpictus mosquitoes in the absence of or much less number of ponds in an area. This expalnation has been added in the revised manuscript (Line no. 592-598 in revised manuscript).

Comment 34: Lines 588-599: not clear why temperature is considered as a major determinant of breeding success, since, as the authors have shown, other factors differ from monsoon and non-monsoon seasons.

Author’s response: Although there are many factors which together are responsible for breeding success of mosquito vectors, among them temperature is considered as one of the major survivibility factors of insects. In the present study, although there is no much differences in water temperature between habitat types during a particular season, but, it was observed that during the winter season, when the temperature of water was very low, the larval population of An. subpictus became decreased to a great extent than monsoon and post-monsoon season. As the mosquitoes are poikilothermic animals, so their internal temperature fluctuates with the temperature of surrounding environment and that is why environmental temperature have great impact in different life history stages of mosquitoes (Line no. 598-605 in revised manuscript). 

Comment 35: Line 603: how much pH, and other abiotic factors, can affect bacterial composition and vice versa? In other words, what is the driving factor affecting other variables considered in this (and other) studies on habitat suitability?

Author’s response: pH of water bodies might have direct or indirect affect on bacterial growth in habitat water. In the present study, physiological tolerance test indicated that, all the four-oviposition attractant bacterial isolates had wide range of pH tolerance (5-11), although they exhibited optimum growth between pH 7.5-9.5 of the culture media. So, when pH of water bodies becomes highly acidic (<5) or highly basic (>11), these bacterial strains would not be able to survive. Anopheles subpictus mosquitoes showed their preference to a pH range between 6.5-7.5 of the habitat water, which is also a tolerable range for bacterial isolates. Physico-chemical parameters of water were co-corelated with one another, such as increase in the dissolve ions and mineral contents of water increased the pH and hardness of water. Hard water with high pH might have negative impact on An. subpictus larvae. So no single factor could play a major role in determining an ideal habitat, rather a group of factors together were responsible for generating a suitable habitat for the breeding and survival of An. subpictus mosquitoes (Line no. 680-687 in revised manuscript).

Comment 36: Lines 613-616: is there the possibility that the contrasting preference between pH in the two species is the result of niche partitioning?

Author’s response: In the present study, larval prevalence of An. subpictus showed negative correlation with the alkalinity of breeding habitat water. According to Ghosh et al. (2020), An. stephensi mosquitoes from some rural and urban areas of West Bengal was also found to have negative correlation with alkalinity of breeding habitat water. But according to the report of Thomas et al. (2016), the same species (An. stephensi) in some areas of south-eastern Chennai showed opposite result, i.e. they showed significant positive correlation with the alkalinity of breeding habitat water. Niche partitioning allows more than one species to live in the same geographical area accessing different resources. But, in this case we recorded the contrasting result in respect to breeding habitat alkalinity preference of An. subpictus with the result as recorded by Thomas et al. (2016). In the present study, among the anopheline larvae, we captured only An. subpictus larvae from the habitat water and no other anopheline species was detected during the entire study period, so there was no chances of niche partitioning. But, if these two species can co-exist in any area and show contrasting preference regarding any physico-chemical parameter (for eg. alkalinity) of habitat water, then it might be concluded that this happen obviously due to niche partitioning.

Comment 37: Line 625: without time-series data it is very speculative to claim possible adaptation to polluted water bodies, as they may just represent cases of oviposition site choice at the boundaries of the species’ preference.

Author’s response: It was merely an observation and this portion has been deleted from the manuscript. 

Comment 38: Line 640: in general, I think that many factors that contribute to larvae abundance and oviposition site choice may be highly correlated one another. You should for example refer to the PCA plots, where you indicated (arrows) which factors co-correlate with the PCA components.

Author’s response: According to the suggestion of hon’ble reviewer, this portion has been described in the revised manuscript (Line no. 364-405 and Tables S6-S9).

Comment 39: Lines 707-709: please discuss the possibility that both Bacillus and Anopheles prefer a certain type of water properties and thus the presence of the former is not a condition but a coincidence with the presence of the latter. (see also next section; and, in general, check this reference: https://www.frontiersin.org/articles/10.3389/fmicb.2019.02036/full - it refers to Aedes species but may contain some useful information for your study)

Author’s response: Earlier reports of several researchers indicated that spore forming Bacillus group of bacteria could have the ability to decompose organic matter of breeding habitat water-bodies, maintenance of phosphate and nitrate contents, hardness of water, pH and alkalinity of water within tolerable limits (Zink et al., 2011; Hainfellner et al, 2018; Hura et al., 2018; Barman et al, 2018; Elsabagh et al, 2018). Bacillus spp. could also enumerate in comparatively more clean, soft and stagnant breeding habitat waterbodies with high dissolve oxygen content, where An. subpictus mosquitoes also showed higher prevalence exhibiting the co-existence of anopheline larvae and spore forming Bacillus bacterial strains. This portion has been discussed in the revised manuscript (Line no. 694-698 in revised manuscript).

Comment 40: Lines 752-761: the tolerance displayed by bacteria may not be relevant for mosquito control if, as the authors showed in this MS, mosquitos have a narrower tolerance towards such abiotic factors.

Author’s response: All the breeding habitat water bacterial isolates were found to tolerate the pH 5-11, although they exhibited optimum growth between pH 7.5-9.5 of the culture media. They could tolerate a temperature range between 150C to 450C, although their optimum growth was recorded at a temperature range between 300C to 350C. These isolates could tolerate up to 4% NaCl concentration of the growth medium. Present observation depicted that higher larval prevalence of An. subpictus at a water temperature ranging between 270C -290C and pH of the habitat water ranging between 6.5-7.5. In these temperature and pH range, breeding habitat bacteria also showed higher amount of growth. (Line no. 784-787 in revised manuscript). 

Comment 41: Lines 776-777: can the authors elaborate a little on which kind of bacterial manipulation can be performed to control mosquitoes?

Author’s response: If these oviposition attractant bacterial isolates could be ruined from the mosquito breeding sites through ecofriendly bactericidal or bacteriostatic plant extracts, the rate of egg laying by mosquito vectors will be minimized, in such a way, which would obviously contribute an alternative strategy of vector management in malaria prone areas. (Line no. 805-809 in revised manuscript).

Reviewer #2: 

Comment 1: STATISTICAL ANALYSIS. The analysis of number of larvae per season in the different ponds have been done using two-way ANOVA. However, giving that the Rice-field are empty during winter, the data are missing (resulting in a non-normal data). This is a strong violation to the ANOVA requirements (there is no standard deviation, and the ANOVA evaluates the variation of the variance). I think a non-parametric test is more appropriate or a different statistical approach.

Author’s response: According to the suggestion of hon’ble reviewer a non-parametric test of two-way ANOVA (Friedman test) have been performed. This portion is corrected in the revised manuscript (Line no. 280-281, 307-309, Table S1).

Comment 2: Analysis of the physio-chemical features on the oviposition sites. The authors used PCA followed by a regression analysis. I do not agree for two reasons: first, apart reporting the r value they should report the associated P. Second: doing that, they are testing multiple variables on the same outcome multiple times and that increase the Type I error in the analysis. I think a more feasible approach is the construction of a model (general linearized or mixed model?) in which all the variables are modelled for their influence on the outcome (the number of larvae).

Author’s response: As per the suggestion of hon’ble reviewer, during correlation analysis, multiple testing correction (Holms-Bonferroni correction) have been performed and the p values associate to the Person correlation coefficients have been given in the revised manuscript (Line no. 289-291, 415-419, Table 3). A generalized linear model (GLM) has been constructed with larval density as dependent variable and all physico-chemical parameters (temperature, dissolved oxygen, alkalinity, pH, turbidity, total dissolved solids, total hardness, electrical conductivity, chloride, nitrate and phosphate) as predictors. This portion has been added in the revised manuscript. (Line no. 291-295 421-431, Table S10 and Fig 7 in revised manuscript).

Comment 3: Student t test of the number of eggs laid using isolates. I do not understand why the authors used an unpaired student t test instead of a paired t. The pairing is the two oviposition cups from the same experiment (those that contained water and isolates).

Author’s response: Yes sir, it is our mistake. We have performed paired t test between test cups and control cups and corrected this portion in the revised manuscript. (Line no. 296, 477-488, Table 8 in revised manuscript).

Comment 4: REPORT OF THE RESULTS. I suggest the authors to remove from the text (and also from the abstract) the description of the results that are already reported in the tables. It does not add anything to the paper, is not easy to read and to interpret and the data are available in the tables.

Author’s response: According to the suggestion of hon’ble reviewer, description of results that are already reported in tables are removed from the text and abstract.

Comment 5: MICROBIOLOGY ANALYSIS. At line 363, in the results. I would avoid calling it "bacteria diversity". There are more sophisticated analysis of diversity and reducing bacteria diversity to the cultivable portion is strongly reductive. Furthermore, I think it is not the focus of the authors. The point here is to find isolates common to all the riding sites that can function ass attractants. I would re-write the paper in this terms, as otherwise is not clear.

Author’s response: According to the suggestion of hon’ble reviewer, the term "bacteria diversity" has been removed in the revised manuscript and written as “Populations of different bacterial groups in breeding habitats”. (Line no. 432 in revised manuscript).

Comment 6: METHODS: Line 87: the authors state that they counted the larvae per dip. How were they sure they are An. subpictus? They mentioned later (in the oviposition assay) that they collected larvae, brought them in the lab and selected An. subpictus for oviposition assays, which means multiple species are present. Did they check all the larvae they've collected?

Author’s response: During larval collection from mosquito breeding habitats, Culex and Aedes larvae were also captured in addition to Anopheles subpictus larvae as the only anopheline mosquito species. When per dip larval density was calculated, only anopheline larvae were counted based on the morphological identification characters like absence of any siphon tube in their last abdominal segment. After emergence of adult from the larvae reared in the laboratory, all of the anophelines were identified and confirmed as Anopheles subpictus by morpho-taxonomic method (Nagpal & Sharma, 1995), and, no other species of Anopheles were identified. Only An. subpictus mosquitoes were sorted for further ovipositional bioassay. This portion has been modified in the revised manuscript (Line no. 179-180 in revised manuscript).

Comment 7: English needs revision. For example, at line 573, "It is very much important"...I would avoid such an hyperbolic phrasing.

Author’s response: According to the suggestion of hon’ble reviewer, the whole manuscript was checked for such type of errors and have been corrected carefully. (Line no. 578-580 in revised manuscript).

---

## [Decision Letter · Decision Letter 1]

16 Jan 2023

PONE-D-22-19958R1Combined effect of physico-chemical and microbial quality of breeding habitat water on oviposition of malarial vector Anopheles subpictus .PLOS ONE

Dear Dr. Chatterjee,

Thank you for submitting your manuscript to PLOS ONE. After careful consideration, we feel that it has merit but does not fully meet PLOS ONE’s publication criteria as it currently stands. Therefore, we invite you to submit a revised version of the manuscript that addresses the points raised during the review process.

Please, take into consideration the stylistic comments from Reviewer 3, to improve the quality of your manuscript.==============================

We look forward to receiving your revised manuscript.

Kind regards,

Cinzia Calvio, PhD

Academic Editor

PLOS ONE

Journal Requirements:

Reviewers' comments:

Reviewer's Responses to Questions

**Comments to the Author**

1. If the authors have adequately addressed your comments raised in a previous round of review and you feel that this manuscript is now acceptable for publication, you may indicate that here to bypass the “Comments to the Author” section, enter your conflict of interest statement in the “Confidential to Editor” section, and submit your "Accept" recommendation.

Reviewer #3: All comments have been addressed

2. Is the manuscript technically sound, and do the data support the conclusions?

Reviewer #3: Yes

3. Has the statistical analysis been performed appropriately and rigorously? 

Reviewer #3: Yes

4. Have the authors made all data underlying the findings in their manuscript fully available?

Reviewer #3: Yes

5. Is the manuscript presented in an intelligible fashion and written in standard English?

Reviewer #3: Yes

6. Review Comments to the Author

Reviewer #3: (No Response)

7. PLOS authors have the option to publish the peer review history of their article (what does this mean?). If published, this will include your full peer review and any attached files.

Reviewer #3: No

---

## [Author Response · Author response to Decision Letter 1]

23 Feb 2023

To

Cinzia Calvio

Academic Editor

PLoS ONE

Sub: Submission of revised manuscript (PONE-D-22-19958R1)

Dear Sir,

Thank you for your email dated 16/01/2023 enclosing the reviewers’ comments. We are very much thankful to you and to the hon’ble reviewers for their valuable comments and constructive suggestions. We have carefully reviewd all the comments and have revised the manuscript accordingly. Our responses are given point-by-point manner below and the modified portions have been highlighted in the revised manuscript. 

We look forward to hearing from you in due course.

With regards,

Dr. Soumendranath Chatterjee

Professor

Department of Zoology

The University of Burdwan

Golapbag, Burdwan, West Bengal

Introduction:

Comment 1: Line 48 – Malarial pathogens, change to “Malaria parasites”.

Author’s response: According to the suggestion of hon’ble reviewer in line 48 the words ‘Malarial pathogens’ have been changed to “Malaria parasites”.

Comment 2: Line 48 – This statement “they can easily transmit this protozoan parasite from one human host to another” is redundant, remove it. The remaining sentence should read “Females of different species of Anopheles mosquitoes serve as vectors of malaria parasites due to their blood sucking behavior”.

Author’s response: According to the suggestion of hon’ble reviewer, the statement “they can easily transmit this protozoan parasite from one human host to another” has been removed from the revised manuscript and written as “Females of different species of Anopheles mosquitoes serve as vectors of malaria parasites due to their blood sucking behavior” (Line no. 47-48).

Comment 3: Line 69 – 72: …where researchers showed….note that, researchers do not show, rather studies finds or show/indicates …..led to reduce .. it should be “led to a reduction in”

Author’s response: According to the suggestion of hon’ble reviewer, this sentence has been modified as “Several studies indicated that killing of these bacteria by sterilization technique or addition of effective antibiotics to the breeding habitat water led to a reduction in ovipositional response by adult gravid female mosquitoes” (Line no. 80-82 in the revised manuscript).

Comment 4: The background has not adequately indicated as to why this study was conducted and what is the significance of the study. Authors should ensure that the rationale of the study is clearly stated in the study background.

Author’s response: According to the suggestion of hon’ble reviewer, background of the study have been described in the revised manuscript (line no. 83-95).

Comment 4: It is also important to slightly describe the malaria burden in the district/area.

Author’s response: According to the suggestion of hon’ble reviewer, malaria burden in the district have been described in the revised manuscript (line no. 58-63).

Methodology

Comment 5: Study area and study period: Based on the nature of the study, it is important to describe some basic climatic features of the study area, including rainfall, temperature, seasons of the year, etc.

Author’s response: According to the suggestion of hon’ble reviewer, climatic features of the study area have been described in the methodology section of the revised manuscript (line no. 102-104).

Comment 6: Field survey & collection of habitat water

Authors have to describe in detail how did they identify and select the breeding sites on which samples were collected. Was it throughout the district, were they randomly sampled, how many were they? 

Author’s response: Samples were collected from four blocks (Tarakeswar, Singur, Chinsurah-Mogra and Panduah) of Hooghly district (mentioned in materials and methodology section under study area and study period; line no. 100-102). In these four blocks of Hooghly district, suspected water bodies were checked randomly for the presence of Anopheles subpictus larvae. Samples were collected from those water bodies, where larval prevalence of An. subpictus were recorded during the study period (line no. 106-108).

Comment 7: Processing of water samples for bacterial isolation 

This subheading should be placed before Analysis of bacterial populations of water 

Author’s response: According to the suggestion of hon’ble reviewer, the subheading “Processing of water samples for bacterial isolation” (line no. 135) was added before “Analysis of bacterial populations of water” 

Results

Comment 8: For clarity and easy comprehension re-arrange table 2 as shown below

Table 2. Seasonwise physico-chemical parameters (Mean±S.E.) of different habitat waterbodies of Anopheles subpictus.

Parameter Summer Monsoon Post-monsoon Winter

 Pond Dain Rice field Pond Dain Rice field Pond Dain Rice field Pond Dain Rice field

Temperature (oC) 

DO (mg/L) 

Alk 

pH 

Turb 

TDS 

T.H 

E.C 

Cl- 

NO3- 

PO4- 

Author’s response: According to the suggestion of hon’ble reviewer, table 2 has been re-arranged in the revised manuscript (Line no. 344-346).

---

## [Editor Report · Decision Letter 2]

24 Feb 2023

Combined effect of physico-chemical and microbial quality of breeding habitat water on oviposition of malarial vector Anopheles subpictus .

PONE-D-22-19958R2

Dear Dr. Chatterjee,

We’re pleased to inform you that your manuscript has been judged scientifically suitable for publication and will be formally accepted for publication once it meets all outstanding technical requirements.

Kind regards,

Cinzia Calvio, PhD

Academic Editor

PLOS ONE
---

## [Editor Report · Acceptance letter]

1 Mar 2023

PONE-D-22-19958R2 

Combined effect of physico-chemical and microbial quality of breeding habitat water on oviposition of malarial vector *Anopheles subpictus*

Dear Dr. Chatterjee:

I'm pleased to inform you that your manuscript has been deemed suitable for publication in PLOS ONE. Congratulations! Your manuscript is now with our production department. 

Kind regards, 

on behalf of

Dr Cinzia Calvio 

Academic Editor

PLOS ONE